# Universal scale-free representations in human visual cortex

Raj Magesh Gauthaman[1]*, Brice Ménard[1,2,3], Michael F. Bonner[1]

**1** Department of Cognitive Science, Johns Hopkins University, Baltimore, Maryland, United States of America, **2** Department of Physics and Astronomy, Johns Hopkins University, Baltimore, Maryland, United States of America, **3** Santa Fe Institute, Santa Fe, New Mexico, United States of America

* rgautha1@jh.edu

## Abstract

How does the human brain encode complex visual information? While previous research has characterized individual dimensions of visual representation in cortex, we still lack a comprehensive understanding of how visual information is organized across the full range of neural population activity. Here, analyzing fMRI responses to natural scenes across multiple individuals, we discover that neural representations in human visual cortex follow a remarkably consistent scale-free organization—their variance decay is consistent with a power-law distribution, detected across four orders of magnitude of latent dimensions. This scale-free structure appears consistently across multiple visual regions and across individuals, suggesting it reflects a fundamental organizing principle of visual processing. Critically, when we align neural responses across individuals using hyperalignment, we find that these representational dimensions are largely shared between people, revealing a universal high-dimensional spectrum of visual information that emerges despite individual differences in brain anatomy and visual experience. Traditional analysis approaches in cognitive neuroscience have focused primarily on a small number of high-variance dimensions, potentially missing crucial aspects of visual representation. Our results demonstrate that visual information is distributed across the full dimensionality of cortical activity in a systematic way, thus revealing a key property of neural coding in visual cortex. These findings suggest that we need to move beyond low-dimensional characterizations to fully understand how the brain represents the visual world.

## Author summary

The human cerebral cortex is thought to encode sensory information in population activity patterns, but the statistical structure of these population codes has yet to be characterized. By examining large-scale neuroimaging recordings of human

**Data availability statement:** The primary dataset analysed in this study is the publicly available Natural Scenes Dataset (Allen et al., 2021), which can be accessed at this URL: https://naturalscenesdataset.org/. We replicate our findings in other publicly available datasets, including the fMRI data from the THINGS-data collection (Hebart et al., 2023) and the THINGS ventral stream spiking dataset (Papale et al., 2025), which can be accessed at the following URLs respectively (https://plus.figshare.com/collections/THINGS-data_A_multimodal_collection_of_large-scale_datasets_for_investigating_object_representations_in_brain_and_behavior/6161151 and https://gin.g-node.org/paolo_papale/TVSD/). All code for these analyses are available at this git repository: https://github.com/BonnerLab/scale-free-visual-cortex.

**Funding:** This research was supported in part by a Johns Hopkins Catalyst Award to MFB, Institute for Data Intensive Engineering and Science Seed Funding to MFB and BM, and grant NSF PHY-2309135 to the Kavli Institute for Theoretical Physics. https://research.jhu.edu/major-initiatives/catalyst-awards/; https://www.idies.jhu.edu/; https://www.nsf.gov/awardsearch/showAward?AWD_ID=2309135&HistoricalAwards=false. The funders did not play any role in the study design, data collection and analysis, decision to publish, or preparation of the manuscript.

vision using a spectral approach more common in physics than neuroscience, we reveal the universal scale-free structure of population codes in visual cortex, which is found in all subjects and at multiple stages of visual processing. Moreover, the underlying dimensions of these scale-free representations are strongly shared across individuals, indicating a remarkable convergence toward a common high-dimensional code, despite differences in visual experience or brain anatomy. These findings reveal high-dimensional aspects of cortical representation that are undetectable with conventional methods, such as representational similarity analysis, and they contradict previous theories suggesting that high-level visual cortex representations are low-dimensional. Together, this work identifies a vast space of uncharted dimensions in the human brain that have been largely overlooked in previous work but may be critical for understanding human vision.

## 1 Introduction

How does the human visual cortex leverage the activity of large-scale neural populations to represent sensory information? To what extent are these visual population codes shared across individuals? We can shed light on these questions by studying the covariance structure of neural populations, a quantity describing important aspects of information processing in neural population activity. While previous studies have examined the covariance of visual cortex activity in humans and monkeys, they have largely done so in the service of dimensionality reduction [1–4]. Some of these studies have argued that dimensionality reduction is implemented by visual cortex itself to achieve stable representations of behaviorally relevant information [5–7]. This theoretical position suggests that the dimensionality of natural image representations decreases across stages of visual processing until cortical activity is constrained to a low-dimensional subspace that drives behavior. Others have used dimensionality reduction as a tool to uncover individual representational dimensions that can be semantically interpreted [1,4,8,9]. Together, these studies have provided valuable insight into the highest-variance dimensions of visual cortex activity. However, none have attempted to provide a *global* characterization of how cortical populations encode sensory information. To understand the sensory code in its entirety, we need to go beyond its leading dimensions and examine stimulus representations at all levels of variance: we need to characterize its *covariance spectrum* as broadly as possible. This question has direct implications for how we understand visual processing: if representations are truly low-dimensional, then current approaches may be sufficient, but if they span thousands of dimensions as suggested by recent findings in mice [10], then previous work may have missed the majority of meaningful cortical information.

It is also crucial to understand which aspects of visual cortex representation are shared across individuals and thus reflect general properties of human vision as opposed to idiosyncratic aspects of each person's neural activity patterns. A number of studies have addressed this question using representational similarity analysis (RSA) and hyperalignment [3,11,12]. However, these previous studies have not

examined the extent to which the underlying dimensions of the cortical sensory code are shared across individuals. Thus, fundamental questions about the nature of population codes in human visual cortex remain unanswered. Over how many dimensions are visual cortex representations expressed, and what is the underlying statistical structure of these representations? Do subjects represent images similarly across all ranks of latent dimensions, or is the shared code restricted to a core subset of dimensions?

Here we explore the structure of visual neural representations following a methodological approach closer to physics than traditional neuroscience. We characterize, as broadly as possible, the spectrum of representational dimensions that co-vary between trials or subjects. We do so by leveraging a large-scale dataset of high-resolution fMRI responses to thousands of natural images [13]. Using an orthogonal cross-decomposition with a generalization test on held-out images, we show that the covariance spectrum of fMRI responses to natural images in human visual cortex is consistent with a power-law distribution over almost four orders of magnitude in the number of latent dimensions. This reveals that the statistics of neural activations are *scale-free* and that the dimensionality of visual cortex representations is *unbounded*, a behavior we observe across all individuals and in multiple visual regions along the cortical hierarchy. These results are in line with recent findings of power-law spectra in the primary visual cortex (V1) of mice [10]. Next, by aligning the activations of different subjects, we show that they share a common representation over many latent dimensions, suggesting that they represent the visual world using a similar scale-free code despite idiosyncrasies in brain anatomy and visual experience. In contrast, we find that conventional approaches in cognitive neuroscience are only sensitive to leading, high-variance dimensions and fail to reveal the full extent of these shared population codes. Together, these findings challenge low-dimensional theories of visual representation and reveal that conventional analytical approaches capture a small fraction of the meaningful information encoded in cortical activity. Our results suggest that fully understanding human vision will require embracing the high-dimensional nature of its neural implementation.

## 2 Results

We set out to study the statistical properties of stimulus-related fluctuations in the fMRI responses of visual cortex and to characterize the spectral properties of visual representations shared across individuals. To do so, we analyze the activation covariance, which captures all second-order (pairwise) interactions between voxels.

An important first step is to characterize the eigenvalue spectrum of this neural covariance, which describes how variance decays across dimensions. The spectral decay informs us about the fraction of dimensions that effectively contribute to the global variance. If the spectrum displays an exponential decay, this indicates that there is a finite effective dimensionality capturing most of the variance. In contrast, if a heavy-tailed distribution is observed, it is necessary to consider the full spectrum because the effective dimensionality may not have an intrinsic upper bound other than the total number of neurons, which is about $4 - 6 \times 10^9$ in human visual cortex [14].

Note that it is possible for individuals to have similar covariance eigenspectra but use different latent dimensions to encode images in distinct ways. Thus, it is necessary to go beyond characterizing the spectrum of stimulus-related variance *within* each individual and identify the degree to which this variance is shared *between* individuals. By directly examining shared variance, we can identify representational properties that are consistent across individuals and are likely to have a fundamental role in visual information processing.

Standard principal component analysis (PCA) of cortical activations is not adequate to reach our goals as it decomposes all sources of variance, including both the stimulus-related signal of interest and noise. In addition, when comparing two individuals, we are considering two covariances expressed in different spaces, which could be rotated relative to one another or contain fundamentally different dimensions. Thus, cross-validation and functional alignment are needed to identify the shared stimulus-related information in cortical activity. Below, we describe a method that generalizes PCA to achieve these goals.

## 2.1 A spectral method to estimate similarities between high-dimensional representations

Given a pair of subjects $X$ and $Y$ who have seen the same images, we compute the covariances of their cortical representations $\Sigma(X, X)$ and $\Sigma(Y, Y)$ as well as their cross-covariance $\Sigma(X, Y)$. Each of these covariance matrices is then decomposed to yield a spectrum of orthogonal latent dimensions with variances $\Sigma_k(\cdot, \cdot)$, where $k$ is the *rank* of the latent dimension when sorted in decreasing order of variance. Each of these latent dimensions is analogous to an eigenvector from PCA and corresponds to a linear combination of voxels in the visual cortex. For each dimension, the pattern of weights over voxels describes the contribution of each voxel to that dimension.

To estimate the level of similarity between the neural representations of both subjects $X$ and $Y$ at each rank $k$, we introduce the correlation function

$$r_k(X, Y) = \frac{\Sigma_k(X, Y)}{\Sigma_k(X, X)^{1/2} \, \Sigma_k(Y, Y)^{1/2}} \,. \tag{1}$$

This correlation function $r_k$ characterizes the range of dimensions over which the representation in one subject is shared with another. Similar to a Pearson correlation coefficient, it normalizes the cross-covariance between two datasets $X$ and $Y$ ($\Sigma_k(X, Y)$ in the numerator) by the geometric mean of their covariances ($\sqrt{\Sigma_k(X, X)\Sigma_k(Y, Y)}$), with the additional advantage that each of these terms is evaluated on held-out test data to prevent overestimation of shared signal.

For this ratio $r_k$ to be meaningful, it is important to estimate the numerator and denominator in the same statistical manner and take into account the following requirements:

1. We want to consider only activations that are stimulus-dependent and discard others. This requires cross-validation of covariances using repeated stimulus trials. For between-subject comparisons, we average over cross-trial comparisons so that the estimator is insensitive to the order of the trials.
2. To characterize the shared dimensions of different subjects, the representations need to be hyperaligned by solving the Procrustes problem, which finds the optimal rotation for aligning two matrices along shared latent dimensions [3].
3. We are interested in results that generalize to new stimuli and do not reflect chance alignments of a particular dataset. To do so, we systematically use cross-validated train/test splits of the stimulus images. This defines an estimator that has a zero expectation value if the activations do not co-vary with input stimuli, in contrast to a standard orthogonal decomposition where covariance estimates are always nonnegative.

We refer to this procedure as cross-decomposition. It uses the orthogonal rotation method from hyperalignment to align representations along a set of shared latent dimensions [3]. A large set of training data is used to learn these orthogonal transformations. Once the representations are aligned, we compute cross-validated covariance estimates along each latent dimension using held-out test data, yielding a cross-covariance singular value spectrum. In summary, the cross-validated singular values from this procedure reflect shared stimulus-related variance that is reliably detected in held-out data. We present a detailed mathematical formalism of these estimators in the Methods section. The main steps are also illustrated in Fig 1.

Finally, we highlight two important considerations for characterizing high-dimensional representations:

1. **A spectral approach** is needed to assess the nature of representations in their full extent and to meaningfully compare high-dimensional quantities. A spectral approach allows us to to determine if representations are localized over a finite range of dimensions or spread over an unbounded range (in practice limited by the dataset), and it allows us to compare individuals across all underlying dimensions of cortical activity. We point out that other approaches for comparing representations, such as RSA and centered kernel alignment, typically extract a single scalar coefficient of similarity that is variance-weighted [11,15]. As a result, they are effectively dominated by a relatively limited number of low-rank dimensions and are not sensitive to potentially many informative high-rank dimensions. We illustrate

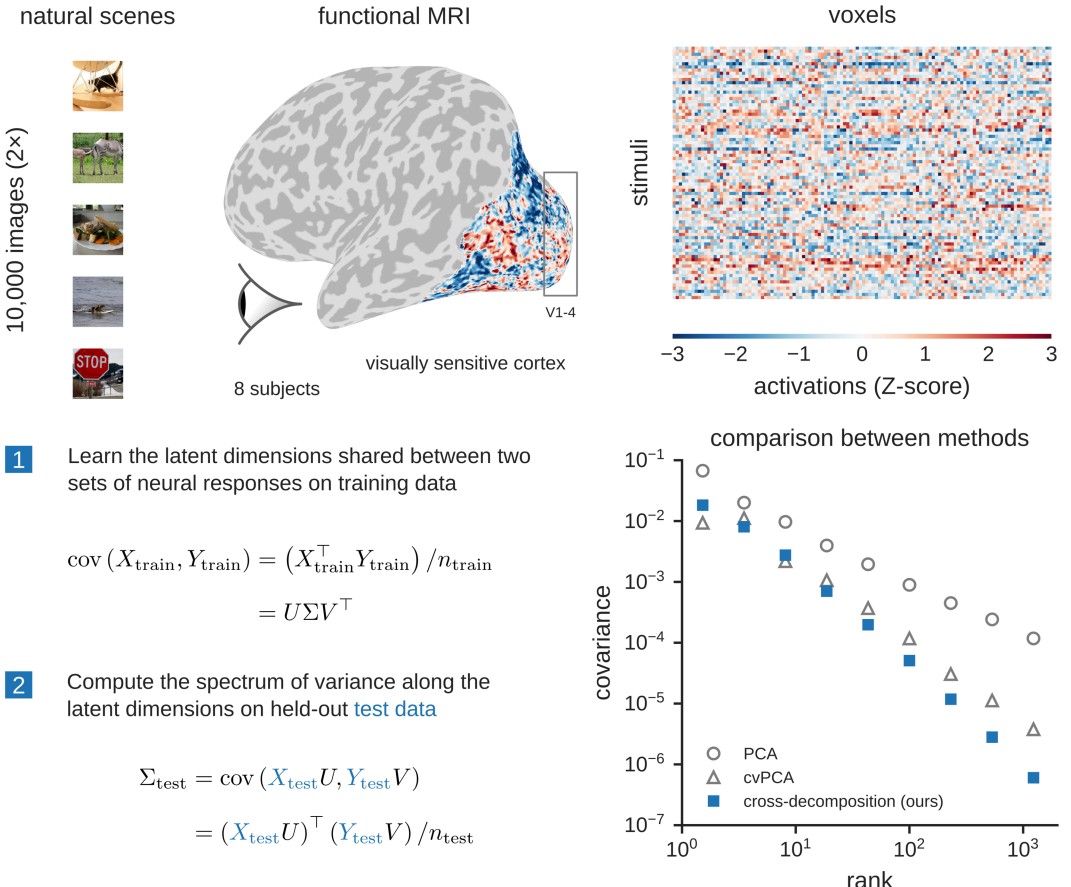

**Fig 1. Cross-decomposition procedure to estimate the cross-validated spectrum of visual representations.** (Top) Each participant viewed about 10,000 natural scene images multiple times while their fMRI BOLD responses were measured. The depicted brain region is a "general" region of interest (ROI) in visual cortex that was identified by selecting all voxels whose activity was modulated by the presentation of images. (Bottom) Estimating a cross-validated covariance spectrum requires identifying directions of variance that are shared between two sets of cortical responses given a set of training images. The two sets of responses can be repeated presentations of stimuli to a single subject or presentations of the same stimuli to two different subjects. The reliable shared variance is then evaluated on cortical responses to held-out test images. Example spectra are shown for subject 1 based on our cross-decomposition approach as well as PCA and cross-validated PCA (cvPCA).

this point in Fig 7, which is discussed in detail in Sect 2.4. To explore high-dimensional properties of neural representations, it is necessary to take a spectral approach and estimate quantities as a function of rank $k$ obtained from an orthogonal decomposition. As described in Eq 1, our goal is therefore to estimate a correlation function $r_k$ rather than a single scalar coefficient.

2. **Aggregating latent dimensions** by binning the statistical estimates in our spectra can provide substantial gains in sensitivity, allowing us to measure spectral shape well beyond the range over which individual components are detected. This comes at the cost of lowering the resolution at which we characterize the latent space of the system, but, if the spectrum presents a smooth dependence on rank, this aggregation can reduce noise with a negligible loss of information. Such an approach is ubiquitous in physics. In contrast, previous work in neuroscience has often focused on latent dimensions that can be individually detected (and, if possible, interpreted). When focusing on individual dimensions, the unavoidable presence of noise imposes severe limitations on the range of dimensions that can be studied and effectively limits such an approach to a low-dimensional view of neural representation. Here, our

PLOS Computational Biology

goal is to probe the statistical structure of activation covariances and characterize the full extent of the spectrum of visual representations and their similarities between individuals.

## 2.2 Scale-free representations in visual cortex

We first used our cross-decomposition approach to characterize, as broadly as possible, the spectrum of image representations in a general region of interest (ROI) including all visually responsive voxels (Fig 2, left). Interestingly, despite focusing on only stimulus-related variance that generalizes to held-out data (i.e., putting strong requirements on the selected variance), we can reliably detect signals following a power-law distribution across nearly four orders of magnitude in the number of latent dimensions. As described in the Methods section, we normalized the overall amplitude of covariances to account for the varying number of voxels between individuals (see S1 Fig). The error bars in Fig 2 depict variance across cross-validation folds, demonstrating that the covariance statistics are reliably estimated.

These findings have several key implications. First, they demonstrate that human visual cortex represents natural images in population activity that is *scale-free*. The scale-free nature of these representations implies that their dimensionality is ill-defined: estimates of effective dimensionality for visual cortex likely reflect the properties of a given dataset such as the number of stimuli or the number of recording channels rather than an intrinsic property of cortical representation. In fact, our findings suggest that the representations of visual cortex are expressed over *all* available dimensions and that, given the power-law behavior, any low-rank truncation of these representations would lead to a non-negligible loss of stimulus-related information. Given that the maximum reachable rank is commensurate with the number of neurons in visual cortex, we expect our measurements to reflect a partial view of the entire neural representation. Second, the observed consistency of these spectra across subjects shows that there is universality in how variance is spread across latent dimensions in different participants. In other words, visual cortex representations have the same level of smoothness across individuals.

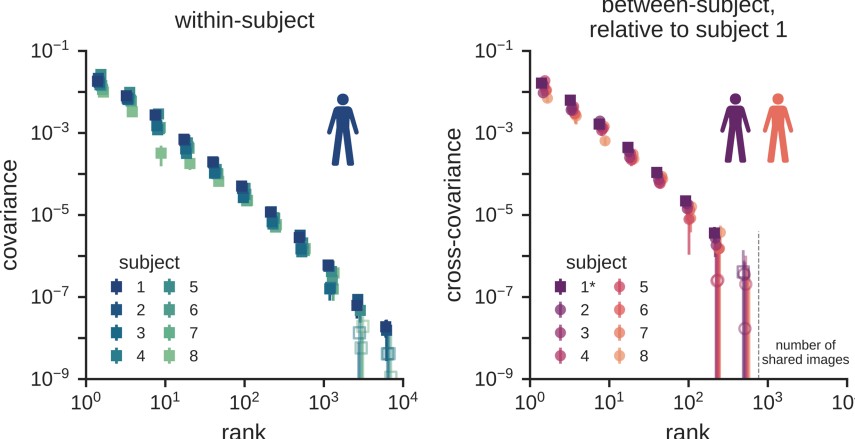

**Fig 2**. **Cortical representations of natural images are scale-free and shared across subjects over many dimensions.** (Left) Within-subject covariance spectra of visual cortex responses in a general region of interest including all visually responsive voxels from the Natural Scenes fMRI dataset [13]. (Right) Between-subject covariance spectra for the same region of interest, characterizing shared variance relative to reference subject 1. The range is limited by the number of shared stimuli seen by all subjects. All spectra are normalized to account for differences in the number of voxels across participants and averaged within bins of exponentially increasing width and across 8 folds of cross-validation. Error bars denote standard deviations across these 8 folds. Fig 3 performs permutation testing to demonstrate the statistical significance of these spectra and S6 Fig shows the results of the same analysis using each subject as the reference subject. Open symbols denote data that are not significant at $p < 0.001$ (permutation tests, $N = 5000$).

To validate the robustness of the measured covariance spectra, we perform null tests by de-correlating the visual stimuli to the recorded brain activation signals. We do so by shuffling the stimulus index when evaluating variance on the held-out stimuli (see Methods). Ten samples of these permuted spectra are shown in the inset of Fig 3, demonstrating that in the absence of reliable stimulus-related signal, our cross-decomposition spectrum is consistent with its expected value of zero. We present the 68th, 95th and 99th percentiles of the empirical null distribution as contours with different shades of gray in the main panel. The actual signals (shown for subject 1) are found to be about an order of magnitude greater than these upper limits, demonstrating their statistical significance over multiple decades of latent dimensions. In addition to this null test, we also perform a analysis involving a non-visual frontal brain region where much weaker covariance signals are expected. As shown in the upper panels of the figure, we indeed find that the covariance estimates for the frontal ROI are about an order of magnitude lower in amplitude than in the "general" visual ROI and are far less reliable in the between-subject analysis (upper right panel). Interestingly, despite the large drop in amplitude, we still observe a similar spectral dependence in the frontal and visual ROIs, suggesting that the scale-free covariance structure of cortical population codes identified here extends beyond visual cortex. Fig 3 also illustrates a power-law fit to the covariance spectra, leading to an index of −1.6 in this subject that is highly consistent across all subjects. However, when attempting to interpret this value, it is important to note that, as illustrated in Fig 1, the observed power-law index depends on the specific statistical estimator used.

We also observed that the typical spatial scales over which fMRI signals fluctuate decrease with rank. This is illustrated in S8 Fig. This suggests that the scale-free property of cortical representations exists over a wide range of physical scales, from tens of centimeters down to the limiting resolution of the experiment (i.e., 1.8 mm). Interestingly, similar scale-free covariance spectra have been reported in the primary visual cortex of the mouse brain [10]. These results

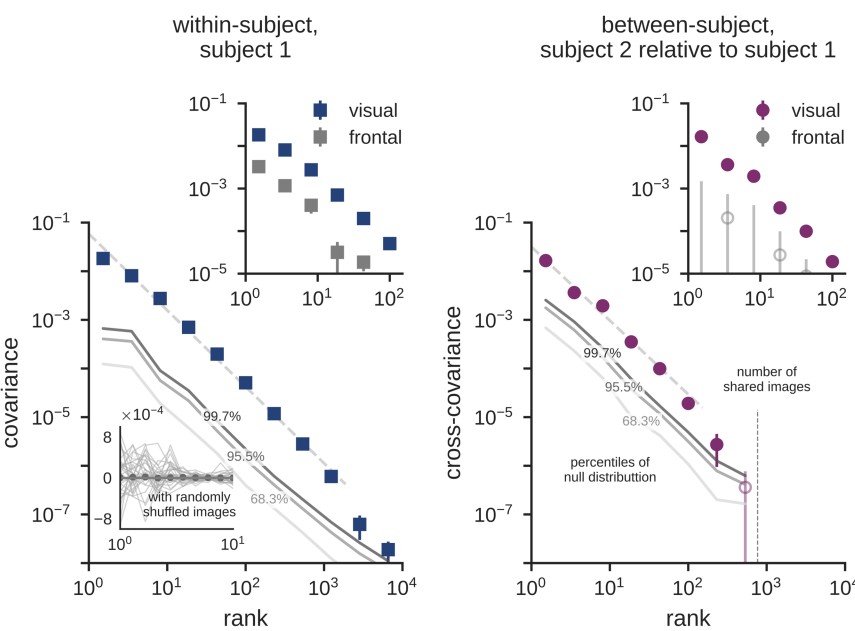

**Fig 3. Null tests and comparison to a non-visual cortical region.** The lower panels compare the activation covariance for subject 1 and the cross-covariance between subjects 1 and 2 (as shown in Fig 2) to spectra obtained for data with randomly shuffled image labels. The 68th, 95th and 99th percentiles of the null distribution, computed with 5,000 random permutations are displayed in gray lines. Ten individual covariance spectra are shown in the inset, along with the mean across all 5,000 permutations (gray dots), which falls to zero, as expected. The upper panels compare within- and between-subject spectra for the visually responsive region of interest (the "general" ROI) to the frontal region. The frontal region displays a signal that is about an order of magnitude lower in the within-subject analysis and much less reliable in the between-subject analysis. Open symbols denote data that are not significant at $p < 0.001$ (permutation tests, $N = 5000$).

are based on calcium imaging measurements of individual neurons (i.e., on physical scales several orders of magnitude smaller than the ones used in our fMRI analysis). Together, our findings and the previous findings in mice point to a possible ubiquity of scale-free activation spectra in mammalian visual cortex. Note that the power-law exponents of the mouse spectra from [10] and the human spectra reported here for the within-subject analysis in Fig 3 cannot be directly compared as they are based on different statistical estimators (see Sect 4.2.4).

In summary, our results show that visual cortex representations are expressed over several orders of magnitude of latent dimensions and display a universal scale-free spectrum. While the variance is dominated by a subspace substantially smaller than the entire space, the scale-free nature of the spectrum suggests that its dimensionality is ill-defined and likely unbounded (up to the number of neurons).

## 2.3 A high-dimensional spectrum shared across individuals

We next explore whether representational dimensions are shared *between* individuals (i.e., whether their latent dimensions co-vary for the same stimuli). Representations can be compared across subjects using methods like RSA [11] or hyperalignment [3], with the latter enabling comparisons of individual dimensions across subjects. This hyperalignment is achieved by our cross-decomposition technique. We use it to estimate the cross-covariance spectrum between individuals $\Sigma_k(X, Y)$ for the subset of 766 images which were shown to all participants at least twice in the Natural Scenes Dataset experiment.

The right panel of Fig 2 shows the results for comparisons relative to subject 1. In this figure, the square symbols represent the within-subject spectrum for subject 1 (as in the left panel), and the circles represent comparisons with other subjects. This analysis reveals that the between-subject spectra are remarkably similar to the within-subject spectrum—again exhibiting a scale-free power-law distribution that spans slightly over two orders of magnitude of latent dimensions. Comparisons between all other pairs of subjects yielded similar results and are shown in S6 Fig. Note that the upper bounds of these spectra are limited by the size of the available dataset—we expect that with more stimuli, even more shared dimensions would likely be revealed. As shown in S2 Fig, reliability diminishes for dimensions near the upper limit determined by the number of stimuli, suggesting that the highest rank dimensions are the most susceptible to noise and misestimation. However, these analyses also show that the reliability of these same ranks can be dramatically improved by simply increasing the number of stimuli. Finally, we also replicated our finding of a high-dimensional shared spectrum in other large-scale datasets, including a different human fMRI dataset (the THINGS dataset of neural responses to object images [16], S11 Fig) as well as a monkey electrophysiology dataset (the THINGS ventral stream spiking dataset [17], S12 Fig), demonstrating that this power-law pattern is neither specific to this particular dataset nor a statistical artifact of fMRI data.

As discussed above, we present null tests and a comparison to the (non-visual) frontal region in Fig 3 for subjects 1 and 2. Fitting the cross-spectrum in the general visually responsive region with a power law over two orders of magnitude leads to an index of about –1.5, similar to the one obtained for the within-subject covariance.

As shown in S3 Fig, if we rely on anatomical alignment alone and assume that subject-specific rotations are irrelevant, we only detect shared variance up to about ten dimensions. Thus, while these dimensions may have similar coarse-scale anatomical properties across individuals, detecting shared dimensions at higher ranks requires subject-specific functional alignment.

In sum, these results show that visual representations co-vary between subjects over many ranks of dimensions, and they suggest that despite differences in experience and cortical anatomy, the visual cortices of different individuals converge to a universal scale-free covariance spectrum of natural image representations, over at least two orders of magnitude.

## 2.4 Similarity of scale-free representations

We next address a central question about the nature of shared representations: To what degree are neural representations shared between individuals? Having performed spectral decomposition of both the within-subject and between-subject covariances, we can now compare the ratio of these covariance estimates to determine the proportion of stimulus-related variance that is shared between individuals. Specifically, we compute the set of correlation coefficients $r_k(X, Y)$ between subjects as a function of rank (Eq 1). As ratios, these correlation coefficients are expected to be noisier than the corresponding covariance spectra. To ensure a sufficiently high signal-to-noise, we aggregated these ratios over wider bins of ranks than those used to compute spectra.

As shown in Fig 4, our analysis reveals a high level of correlation over a range of latent dimensions, a finding that is consistent for all subjects considered in the analysis. While we can only detect the correlations among subjects over two orders of magnitude in ranks, our observation of universal power-law distributions in both the within-subject and between-subject spectra suggests that this correlation likely holds well beyond this range (Fig 2). This prediction can be tested in the future with larger datasets. It is also worth noting that while these plots show a remarkable degree of similarity across subjects, some level of subject-specific stimulus-related variance gradually emerges with ranks. An intriguing question for future work is to investigate whether this subject-specific variance reflects meaningful individual differences in visual processing and to determine whether individual differences reliably increase as a function of latent dimension rank.

Our cross-decomposition approach together with a spectral analysis were critical for revealing the high-dimensional nature of these shared representations. Specifically, to meaningfully compare population activity between subjects, it is necessary to *functionally* align their representations into a shared space by applying hyperlignment with subject-specific rotation matrices [3]. If we use anatomical alignment alone and assume that subject-specific rotation matrices are not needed, we only detect representational similarity for the first ten dimensions, as shown by the triangle symbols in Fig 4. In addition, it is important to note that variance-weighted estimators without any spectral decomposition are mainly sensitive to low-rank dimensions and therefore not adequate to probe the properties of high-dimensional representations. We

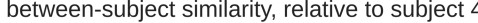

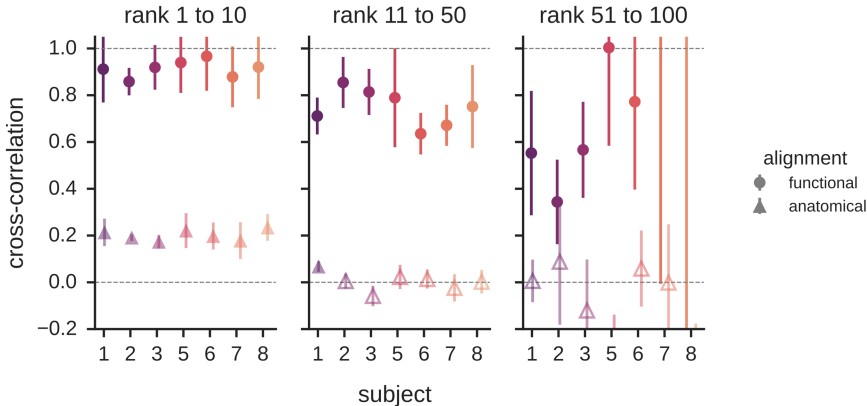

**Fig 4**. **Representational similarity in visual cortex for pairs of subjects.** Correlation of neural representations as a function of ranks obtained from the ratio of cross-validated between-subject variance relative to cross-validated within-subject variance (Eq 1). These findings show that the stimulus-related variance in cortical activity is largely shared across subjects over many ranks. Furthermore, the similarity of high-rank dimensions can only be detected when the representations are functionally aligned in a shared space using subject-specific rotation matrices. With anatomical alignment alone, shared representations are undetectable beyond ≈ 10 dimensions. Error bars denote standard deviations across 8 folds of cross-validation. Open symbols denote points where the mean is less than three standard deviations above zero. S7 Fig shows the results of the same analysis using each subject as the reference subject.

found that a conventional implementation of RSA is effectively sensitive to only a handful of latent dimensions and cannot distinguish between low- and high-dimensional representations, as demonstrated in Fig 7. Furthermore, visualizations show that the low-rank dimensions detected with anatomical alignment and RSA correspond to coarse-scale gradients of cortical tuning, as shown in S8 Fig. Thus, the conventional methods of cognitive neuroscience provide a view of neural representation that is effectively low-dimensional and spatially coarse.

In sum, these findings show that the stimulus representations of visual cortex are not only scale-free but also encode similar information in the brains of different individuals. Most of these representational dimensions have subject-specific anatomical properties, making them undetectable with anatomical alignment alone, and they are obscured in the representational similarity matrices of RSA—functional alignment and spectral analyses are necessary for revealing these shared dimensions. The shared nature of these representations implies that the entire spectrum of activity contributes to the sensory code. This suggests that there is a wealth of representational content in measurements of visual cortex that has remained beyond the reach of conventional methods but may be crucial for understanding human vision.

### 2.5 Similar behavior across visual cortex regions

After focusing on a large and general ROI of visually responsive voxels, we next investigate whether our findings are representative of the structure of population codes observed across different visual regions. To answer this question, we first compute both the within- and between-subject cross-decomposition spectra for the retinotopic regions V1, V2, V3, and V4. As shown in Fig 5, we find the spectra in these regions to be consistent between subjects and similar to the spectra observed for the general ROI, despite these retinotopic ROIs being much smaller in size. This is observed for both the within-subject and between-subject cross-decomposition analyses. This finding suggests that the activation covariance structure is similar over a range of spatial scales. Additionally, we demonstrate using a Gabor filter bank that the observed spectra in these regions cannot be fully explained by basic receptive field properties alone (S4 Fig), replicating previous findings in mouse V1 [10].

We next perform direct comparisons of V1 to V4. First, overlaying the spectra for these regions reveals a trend for a slight decrease in the slope from V1 to V4 (Fig 6). This suggests that although all these regions express their representations in high dimensions (here detected over three orders of magnitude), there may nonetheless be subtle changes in the covariance structure across processing stages. We leave the study of these subtle changes for future work. Here, we demonstrate that our cross-decomposition approach can be used to quantify representational transformations from V1 to V4 by computing *cross*-covariances between different brain regions. To do so, we compute the spectrum of reliable stimulus-related variance shared between each pair of ROIs on different trials within a subject (S5 Fig). We summarize all these pairwise covariances in Fig 6. These plots show that the amount of shared variance decreases along each row/column, with neighboring regions sharing more information than distant regions, as expected based on the successive non-linear transformations that occur along the visual hierarchy. Importantly, all these patterns of variations are not limited to a fixed number of dimensions. They are detected over three orders of magnitude in ranks, showing that these visual transformations are expressed in high-dimensional spaces. To further validate our measurements, we also show a comparison with a frontal region. As expected, this region has weak covariance with visual regions; however, it is interesting to note that even though the covariance between frontal and visual regions is low, there is nonetheless an increase from V1 to V4 at all three decades of ranks.

## 3 Discussion

We discovered that human visual cortex encodes natural scenes through population activity with a universal scale-free structure—a power-law distribution of variance that extends across nearly four orders of magnitude of neural dimensions. This scale-free organization appears consistently across all individuals, multiple visual regions, and different spatial scales, revealing a fundamental principle of cortical representation. Most remarkably, when we align neural responses

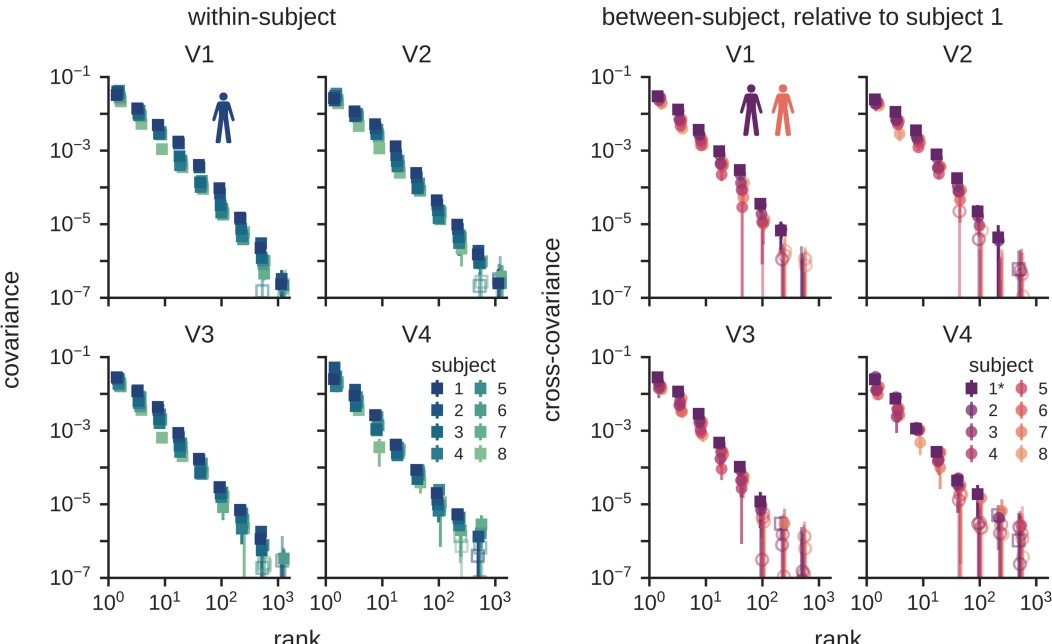

**Fig 5. Scale-free representations are consistently observed across multiple regions of visual cortex.** These plots show within-subject (top) and between-subject (bottom) covariance spectra for retinotopic regions along the visual hierarchy from V1 to V4. In comparison with the larger region of interest shown in Fig 2, these findings demonstrate that the same scale-free power-law distribution is observed in smaller regions of interest, and they show that these power-law distributions are highly consistent across regions. This implies that across these regions of the visual hierarchy, there is universality in the smoothness of cortical population codes. Further, the between-subject spectra show that each of these regions represents images using high-dimensional codes that are shared across individuals. Error bars denote standard deviations across 8 folds of cross-validation. Open symbols denote data that are not significant at $p < 0.01$ (permutation tests, $N = 5000$).

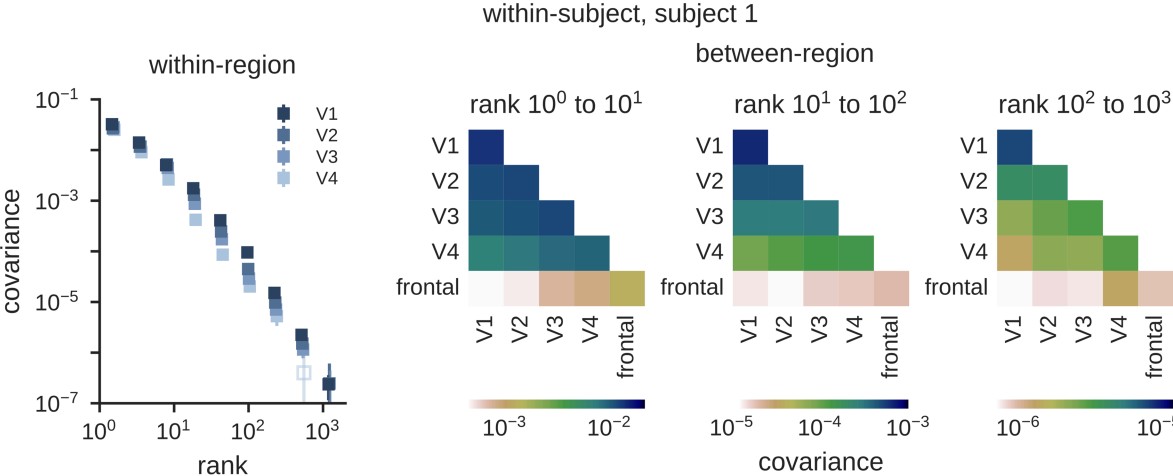

**Fig 6. Comparison of activations between different brain regions.** (Left) Activation covariance spectra for visual cortex regions V1 to V4 in subject 1. (Right) Activation cross-covariance spectra computed for between-region comparisons of visual and frontal regions. Each cell represents the covariance shared between a pair of regions across different trials within subject 1 for a decade of ranks. The amount of shared variance decreases along each row/column reflecting the gradual non-linear transformations that occur during visual processing. As expected, covariances between visual and frontal regions are systematically much weaker.

across individuals using hyperalignment, we find that these representational dimensions are not just similar in their statistical structure but are actually shared between people—individuals encode visual information using the same high-dimensional representational space despite differences in brain anatomy and visual experience. These findings overturn the prevailing view that neural representations can be adequately characterized by a small number of high-variance dimensions. Instead, visual information is distributed across the full spectrum of latent dimensions in cortical activity. This suggests that the brain's visual code is fundamentally high-dimensional and that traditional neuroscience methods, which focus on only the most prominent dimensions, have been missing the vast majority of stimulus-related information encoded in visual cortex.

### 3.1 Implications for neural mechanisms

The observation of consistent spectral behavior across individuals, brain regions, spatial scales, and possibly species is a strong indication that a simple, generic neural mechanism is at play, likely to arise from principles of network self-organization. It is important to understand its nature as it speaks to the fundamental principles that govern the activity and computations of neural populations. Previous work suggests several possible hypotheses, which are not mutually exclusive. It has been proposed that power-law covariance eigenspectra emerge in neural populations that strike an optimal balance between expressivity and robustness [10]. Along these lines, [18] showed that when the covariance eigenspectra of artificial neural networks are made to achieve this balance through regularization during learning, they exhibit a better representational match to human visual cortex. Using connectivity arguments, [19] showed that a Hebbian learning mechanism yields heavy-tailed neuronal connectivity distributions which, under certain conditions, may lead to scale-free covariance spectra. One appeal of such a Hebbian learning account is that it does not require goal-directed feedback (i.e., backpropagation). An exciting direction for future work is to identify which principles govern the underlying neurobiological mechanisms leading to the scale-free representations of mammalian cerebral cortex.

Previous work has argued that the level of dimensionality in a brain region is linked to the region's functional specialization, with some functions calling for high dimensionality and others low dimensionality [e.g. [7,20]]. In the case of vision, there have been conflicting proposals about the link between dimensionality and visual information processing. Some have argued for the importance of dimensionality *reduction* along the visual hierarchy to suppress irrelevant variance and yield robust representations of behaviorally relevant variables [6,7,21,22]. Others have argued for the importance of dimensionality *expansion* to improve linear separability of stimulus classes [23,24]. Our findings show that natural image representations are consistently expressed over a wide range of dimensions in all visual regions examined, suggesting that dimensionality is remarkably stable and neither compresses nor expands as representations are transformed along successive stages of the visual hierarchy. This aligns with recent results demonstrating high-dimensional neural codes throughout the cortical hierarchy in mice performing a decision-making task [25]. However, note that we observed a slight decrease in the slope of the covariance spectra from V1 to V4, suggesting that there may be subtle changes in the covariance structure across these regions. Exploring this trend will be the subject of future work.

We note that this high-dimensional structure is observed when analyzing a rich, large-scale dataset containing neural responses to many thousands of natural images sampling a diverse range of scenes and objects. It is unclear to what extent this power-law covariance spectrum in neural responses depends on the diversity of the stimulus set. In mouse V1, even fairly simple synthetic stimuli still evoked high-dimensional responses (Figure 3 in [10]). Whether a less rich stimulus set would evoke the same power-law spectra in high-level human visual cortex remains an open question. Another important question for future work is whether substantial differences in dimensionality may emerge in downstream regions that support higher-level visual cognition. Recent theoretical and empirical evidence suggests that complex naturalistic behaviors and their associated cortical activity patterns may be more high-dimensional than previously assumed based on evidence from tightly controlled experimental tasks [25–28].

Prior investigations have shown that visual cortex representations are at least partly shared across people and species [3,11] and that hyperalignment is critical for uncovering the common dimensions that underlie these shared representations [3,12]. Our work pushes this exploration further by showing that the shared representations are scale-free: they are expressed over all dimensions and only limited by the size of the data. Thus, despite differences in experience and brain anatomy, the human visual system appears to converge to a common high-dimensional representation of natural images (Fig 2, right).

As in previous work [3], our findings show that the shared representational dimensions of different brains can only be fully observed by hyperaligning the cortical activity patterns from one individual to another. This implies that the underlying dimensions of the representational code may not be spatially localized but instead correspond to latent dimensions that are distributed. Although the brain clearly exhibits spatially localized functions on large scales (e.g., the visual cortex is well localized), our results indicate that when examining brain activity on progressively smaller scales, the representations become more distributed. Indeed, we only found evidence for shared spatial localization in the first decade of the representational spectrum, with the bulk of stimulus-related information expressed through universal latent dimensions that have distinct spatial distributions in each subject and require subject-specific rotation matrices. However, we note that the anatomical alignment procedure used to align individuals to the standardized MNI space may not be optimal. More precise cortical alignment techniques based on cortical folding, function, and other anatomical properties such as cytoarchitectonics could potentially allow for the detection of more shared dimensions across subjects. An important goal for future work is to understand what kind of learning mechanism can explain how the human brain converges to a shared representation over many dimensions despite subject-specific spatial organization. Some insight might come from studies of artificial neural networks, which also exhibit universal learning properties that can be detected in their latent dimensions rather than their neurons [e.g. [29,30]]. Interestingly, the power-law index of neural network representations has recently been used as a metric of representation quality, suggesting that natural and artificial visual systems share similar global population statistics [31].

Previous work has shown that other aspects of neural activity are also characterized by scale-free structure, with much of this previous work focused on the temporal domain, in contrast with the spatial domain examined here. Scale-free structure has been observed in temporal patterns of neural activity examined over a wide range of scales: from the rapidly fluctuating activity of individual neurons up to the slow dynamics of the fMRI BOLD signal [32–36]. The connection between our results showing scale-free properties of activation covariance and previously characterized scale-free properties in the temporal dynamics of neural activity remains an exciting open question for future work.

## 3.2 Limitations of our analysis

### 3.2.1 Linear methods.
In this work, we explored the properties of neural representations using linear methods, based on an orthogonal decomposition of variance in all directions, with a generalization test in held-out data. This linear description of activations is appealing as it characterizes the information that could be decoded by downstream neurons implementing a simple linear readout procedure, which is a key motivation for the utility of this approach. However, it is possible that nonlinear mappings of representations could provide lower-dimensional descriptions of visual cortex activity [21,37,38]. If so, the shared representation between individuals could potentially be an intrinsically lower dimensional manifold embedded in a higher dimensional linear space. An exciting direction for future work is the pursuit of cross-validated nonlinear dimensionality methods that can allow for direct comparisons of linear and nonlinear dimensionality measurements of stimulus-dependent variance.

### 3.2.2 Alternative heavy-tailed distributions.
While our results are consistent with scale-free neural data that obey a power law over many orders of magnitude—as observed from the linearity of the spectra on the log-log plots—there may be other heavy-tailed distributions that are also compatible with these data, such as lognormal distributions, or power-law distributions with an exponential cut-off. We remain agnostic to this possibility, but we emphasize that the unbounded

scaling of our spectra with dataset size (S2 Fig) suggests that we should not attempt to quantify the dimensionality with a single scalar value: there is reliable visual information distributed over all latent dimensions, up to the limit of detectability. Successfully distinguishing between multiple heavy-tailed distributions requires collecting enormous datasets to achieve high enough spectral resolution, especially with the binning procedure we use to extract signal in noisy subspaces. Indeed, similar results in the systems neuroscience literature also refer to such high-dimensional spectra as power-law spectra, though in principle other heavy-tailed distributions may also fit the data reasonably well [10,39].

### 3.3 Comments on traditional approaches in cognitive neuroscience

**3.3.1 The "low-resolution" view of variance-weighted estimators in neuroscience.** The scale-free nature of sensory representations observed here has important implications for the methods of neuroscience. First, it is a common practice to examine neural representations through dimensionality reduction [1,3–5,40]. In such approaches, researchers typically invoke a criterion to identify a subset of "signal" dimensions that account for much of the variance in the data, while discarding the remaining dimensions, which are considered either unimportant or noise-dominated. However, a robust estimate of the covariance spectrum reveals a power-law distribution rather than a distribution with an exponential cutoff. This implies that there is no intrinsic upper bound on the "signal" dimensions, and it suggests that while typical dimensionality-reduction methods may account for much of the *variance*, they discard much of the *information*, which can be found in the long tail of low-variance but nonetheless reliable dimensions. Another important implication is that typical *variance-weighted* estimators, including RSA, voxelwise encoding models (i.e., regression), and linear classifiers, are insensitive to the full extent of information in a power-law distribution. Instead, these methods exhibit a sensitivity to latent dimensions that decays rapidly with rank and becomes vanishingly small beyond the first decade of ranked dimensions. Thus, the correlation metrics obtained with these methods are unable to fully capture the richness of representations expressed in high dimensions. We demonstrate this point in Fig 7 by presenting RSA analyses of representations in low and high dimensions, which show that this technique is effectively insensitive to the effects from ranks greater than about 10. Moreover, because low-rank dimensions correspond to large-scale cortical gradients, this means that typical variance-weighted methods, like RSA, effectively view neural representations through a spatially low-resolution lens. This suggests that there is a vast space of uncharted dimensions in neural activity that has been out of reach for conventional methods and whose role in human cognition has yet to be thoroughly investigated. Future methodological innovations in cognitive neuroscience are required to make further progress elucidating the function of these latent dimensions—perhaps involving information-theoretic estimators that are not variance-weighted. While methods that do not apply variance-weighting, such as canonical correlation analysis (CCA), are sensitive to low-variance but potentially meaningful latent components, they may also detect spurious alignments [15], necessitating the use of robust cross-validation to ensure that only reliable signal is detected.

**3.3.2 Limitations of brain maps and semantic interpretations.** Another consideration raised by our findings is that common methods used for visualizing and interpreting neural representations are severely limited in their ability to characterize high-dimensional information. One such method is brain mapping, which cognitive and systems neuroscientists use to characterize the spatial organization of response preferences to representational dimensions [2,4,16,41,42]. One brain map can only convey a small number of representational dimensions. However, as we have shown, human brain representations are expressed using the full dimensionality of the available space (ultimately defined by the number of voxels) and through a scale-free activation spectrum. An unrealistic number of brain maps would be needed to convey the robust high-dimensional sensory information that our spectral analysis reveals. As a result, brain map visualizations can only scratch the surface of neural representations and cannot illuminate the full extent of the reliable information in cortical population activity. Furthermore, because of the need to hyperalign subjects to find their shared dimensions, the common

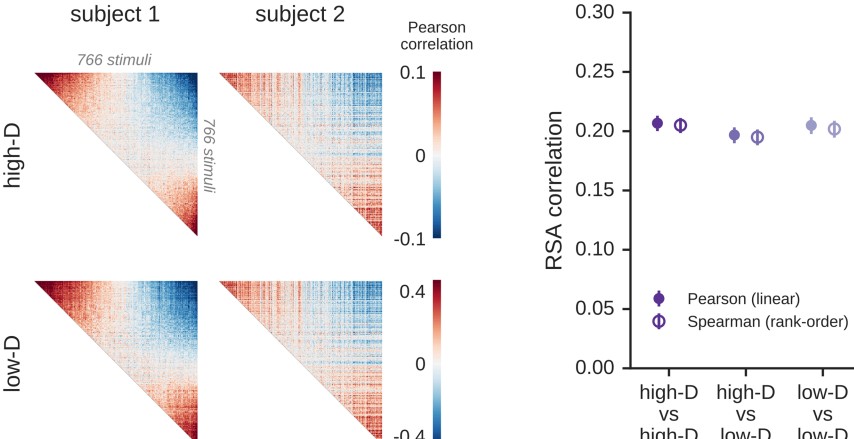

**Fig 7**. **RSA correlations are largely insensitive to high-dimensional structure.** Representational similarity analysis (RSA) was used to compare the visual cortex representations of two individuals. (Left) This analysis was performed on two sets of representational similarity matrices (RSMs): one computed on the original fMRI data ("high-D") and a second computed on reduced-rank fMRI data ("low-D", from the first 10 principal components of the data). Similarities between stimuli were computed using Pearson correlations though similar results were obtained when computing RDMs using the Euclidean metric. As can be seen by the RSM visualizations, this dimension-reduction procedure has little effect on the RSMs because the underlying similarity metrics are largely driven by the leading PCs of the response matrices. (Right) This panel shows the RSA correlation for this pair of subjects, which remains nearly identical even after the data for one subject in each pair has been drastically reduced to just 10 PCs (compare "high-D vs high-D" to "high-D vs low-D"). The same phenomenon is observed when computing RSA scores using both linear Pearson correlations (closed circles) and rank-order Spearman correlations (open circles). These results illustrate that variance-weighted similarity metrics, which are dominated by the leading PCs of neural activity, are effectively insensitive to the shared variance beyond the first 10 ranks. In contrast, the shared variance detected with the cross-decompositions approach extends across multiple decades of ranks (Fig 3). Error bars denote 2 standard deviations across 5,000 bootstrap samples.

approach of averaging fMRI results over anatomically aligned brains effectively limits our view to the relatively small number of dimensions whose anatomical distributions are highly similar across subjects. We suggest that understanding neural representations in their entirety requires a transition from a localized, map-based view of the brain to a more distributed statistical description that considers the full spectrum of the cortical code.

A similar consideration applies to the semantic interpretation of representational dimensions, which is typically restricted to low-rank dimensions that can be individually detected without the need for spectral binning [1,2,4,16]. For example, we find that the first latent dimension in the "general" region of interest correlates strongly with the presence of people (S10 Fig), and more generally, that the first few latent dimensions contain information about the presence of people, food and text in an image (S13 Fig). Additionally, lower-rank components from the THINGS fMRI dataset correlate with multiple interpretable properties of these images (S14 Fig). However, we caution that any semantic meaning we assign to these dimensions may be confounded by low-, mid- or even high-level visual features due to the statistics of the specific dataset used. For example, in S9 Fig, we observe that the 2nd latent dimension of V1 responses appears sensitive to airplanes/trains, the 3rd latent dimension is activated strongly by flowers in vases, and the 4th is activated by clocktower-like buildings. However, we do not expect V1 to represent these semantic properties of images; rather, it is likely sensitive to low-level features, such as different spatial orientations, that covary strongly with these semantic features in this dataset.

As we showed, aggregating dimensions allows one to dramatically extend the range over which latent dimensions are detectable. Such a binning procedure, though common in other fields that examine power-law distributions [43], has not typically been used in studies of neural representation. Doing so requires that we focus on the spectrum as a whole rather than seeking to interpret each latent dimension. Intriguingly, scale-free properties can be found in multiple aspects of the

brain, as previous studies have detected scale-free power-law distributions in temporal activity patterns, structural connectivity, and behavior [19,36,44]. This suggests that methods for rigorously characterizing phenomena distributed over many orders of magnitude may prove crucial to understanding the human brain. To be clear, we believe that the visualization methods of previous work have revealed important organizing properties of the brain. However, if we want to move beyond low-dimensional characterizations of neural representation, we need to embrace approaches for exploring representations in their full extent and understanding them mathematically rather than visually or semantically [45].

Understanding what kinds of visual information are captured by higher dimensions of cortical activity remains an interesting open question. We predict that the most promising approach will be to understand these dimensions in aggregate without attempting to describe each individual dimension. Our work suggests specific hypotheses that could be explored in future experiments. First, it is important to note that the nature of these higher dimensions is expected to differ across the cortical hierarchy. In early visual regions, we predict that higher dimensions encode fine-grained image statistics, including high-resolution features and subtle differences in orientations, spatial frequencies, textures and so on. In contrast, we predict that in later visual regions, higher dimensions encode fine-grained conceptual and geometric properties, such as object identity, materials, 3D shape, the compositional structure of a scene, and so on. We can see parallels of this idea in analyses of generative neural networks, which show that the PCs of latent codes in these networks represent increasingly finer aspects as a function of rank, with lower-rank PCs encoding coarse semantic properties, such as basic-level object categories, and higher-rank PCs encoding finer details, such as the shape and material of objects and the composition of the scene [46].

### 3.4 Summary

Together, our findings reveal the scale-free format of the cortical code for human vision. We found that natural images evoke reliable variance with a power-law spectrum across multiple orders of magnitude of latent dimensions. Importantly, we also found that a large number of these dimensions are shared across individuals, suggesting a remarkable degree of convergence despite individual differences in neuroanatomy and experience. To obtain a global view that captures more than the first ~10 dimensions, it is crucial to use both spectral analysis and functional alignment. Otherwise, one is typically limited to high-variance dimensions (as is the case for RSA) or the subset of dimensions that tend to be anatomically aligned. In sum, these findings suggest that the sensory code of visual cortex spans all available dimensions of population activity and that fully understanding human brain representations requires a high-dimensional statistical approach.

## 4 Methods

### 4.1 Dataset

**4.1.1 Experimental design.** We used the Natural Scenes Dataset (NSD), a large-scale 7T fMRI dataset of image-evoked blood-oxygen-level-dependent (BOLD) responses to approximately 10,000 stimuli in each of eight participants. The stimuli were taken from the Microsoft Common Objects in Context image dataset [47], and they contain photographs of a large and diverse set of objects in their natural scene contexts. Participants viewed these images in the scanner while performing a recognition memory task (i.e., "Have you seen this image before?"). Images were shown three times over the course of the experiment, and trial-level response estimates were obtained for ~30,000 stimulus presentations (about 10,000 images × 3 trials per image). For simplicity, we report results only on the first two presentations for all analyses though they are consistent when analyzing different pairs of presentations. A subset of 1,000 images were seen by all participants in the experiment while the remaining images were unique to each participant. However, note that some participants did not complete all scan sessions and, thus, had fewer stimulus trials. As a result, we used the subset of 766 images that were seen at least twice by all participants for all between-subject analyses, even if some pairs of subjects may have more images in common. This ensures that comparisons between all pairs of subjects are fair and rely on the

same underlying stimulus set, with the tradeoff that we do not use all the available data and thereby reduce signal-to-noise.

**4.1.2 Data preprocessing.** We used the 1.8 mm volumetric preparation of the data, with version b2 of the betas (betas_fithrf), which were z-scored within each scanning session to reduce non-stationarity, as recommended by the authors. We refrained from using the denoised version b3 of the betas (betas_fithrf_GLMdenoiseRR) since (i) the dimensionality reduction applied by the denoising might remove reliable low-variance signal and (ii) these betas were specifically optimized to maximize reliability across trials, which could bias our analyses of reliable cross-trial variance.

**4.1.3 Regions of interest.** For our main analyses, we focused on a pre-defined "general" region of interest (ROI), which included all stimulus-responsive voxels in visual cortex (i.e., the large "nsdgeneral" ROI ranging in size from 12,000 to 18,000 voxels across subjects). We also performed follow-up analyses in smaller ROIs (i.e., the retinotopic regions V1, V2, V3, and V4) to explore possible changes along the visual processing stream. Additionally, we defined a large frontal region comprising voxels that are not strongly modulated by visual stimuli. Specifically, this ROI does not overlap with the "general" ROI and is approximately matched with it on the number of voxels.

## 4.2 General formalism for cross-decomposition

Given the representations of $n$ stimuli in two different neural systems, the neural responses can be organized into two data matrices $X \in \mathbb{R}^{n \times d_X}$ and $Y \in \mathbb{R}^{n \times d_Y}$ where the rows correspond to $n$ presented stimuli and the $d_X, d_Y$ columns correspond to the numbers of recording channels — here, voxels from functional neuroimaging.

Our goal is to characterize the spectrum of reliable variance shared between the two systems $X$ and $Y$. To ensure that this variance generalizes to novel stimuli, we compute the spectrum in a cross-validated manner. Specifically, we split $X$ and $Y$ into *training* and *test* sets $X_{\text{train}}$, $Y_{\text{train}}$ and $X_{\text{test}}$, $Y_{\text{test}}$. All the data are centered using the mean of the training sets.

Our first step is to compute the singular value decomposition (SVD) of the cross-covariance of the training data:

$$\text{cov}(X_{\text{train}}, Y_{\text{train}}) = \frac{1}{n} X_{\text{train}}^{\top} Y_{\text{train}} = U \Sigma_{\text{train}} V^{\top} .$$

This produces two orthonormal matrices $U$ and $V$, whose columns are the left and right singular vectors. $U$ and $V$ are rotation operators that project data from $X$ and $Y$ into their shared latent space, while $\Sigma_{\text{train}}$ is a diagonal matrix whose elements, the singular values, represent the variance shared between $X$ and $Y$ along each latent dimension. However, these singular values are always nonnegative and represent all the variance in the data, whether stimulus-related or not.

Our second step is to evaluate the *reliable* variance along each of these latent dimensions on the held-out test set by projecting the test data onto the shared latent space using the rotation matrices $U$ and $V$ and evaluating their covariance in the latent space:

$$\Sigma(X, Y) = \text{cov}(X_{\text{test}} U, Y_{\text{test}} V) .$$

When the train and test samples are drawn from the same distribution, the covariance matrix $\Sigma(X, Y)$ is diagonal in expectation. In practice, under normal circumstances, $\Sigma(X, Y)$ is quasi-diagonal and we have verified that the contribution from off-diagonal terms is negligible. We thus estimate the cross-validated spectrum as

$$\hat{\Sigma}_k(X, Y) = \text{diagonal of } \Sigma(X, Y) ,$$

where $k$ denotes the rank-dependence of the spectrum.

Note that for all of these spectral analyses, the test singular values $\hat{\Sigma}_k(X, Y)$ are not guaranteed to be positive, unlike a typical singular value spectrum. In fact, if the two systems share no reliable covariance that generalizes from the training set to the test set along a particular singular vector, the expected value of the corresponding test singular value is 0. We make use of this property to perform the null test shown in the inset of Fig 3.

**4.2.1 Cross-validation scheme.** To use all available data, we divide the stimuli into 8 folds, then use 7 folds to learn singular vectors and the remaining fold to estimate variance along the dimensions. Specifically, $X_{\text{train}}$ has 7 times as many stimuli as $X_{\text{test}}$. We repeat this process 8 times using each fold as the test set once, producing eight spectra. To increase signal-to-noise at high-ranks, we average each of these spectra across ranks within bins of exponentially increasing width. To ensure consistency, we used the same bins across all spectra shown in the manuscript, even if the number of stimuli/voxels varies across subjects. These logarithmically spaced bins were constructed by geometrically sampling the maximum range of possible ranks (1 to 10,000) with an approximate density of 3 bins per decade, corresponding to 11 total bins across these 4 orders of magnitude. This density was chosen as a tradeoff between maximizing the spectral resolution by increasing the number of bins and maximizing the signal-to-noise ratio in high-rank bins by averaging over a larger number of ranks. Our results are qualitatively similar when using different numbers of bins. Finally, we average the binned spectra across all 8 folds to obtain $\hat{\Sigma}_k(X, Y)$.

**4.2.2 Normalization of spectra.** The amplitude of the covariance spectrum depends on the number of channels $d_X$ and $d_Y$ in the data. Specifically, since the responses of all voxels are independently z-scored using their means and standard deviations across training images, each voxel has approximately unit variance across the entire dataset, making the total variance of the dataset approximately equal to the number of voxels. To compensate for trivial variations in the spectra caused by comparing representations in regions of different sizes, we divide all spectra by the geometric mean $\sqrt{d_X d_Y}$. This normalization procedure ensures that the covariance spectra cumulatively sum to 1 on the training data, and would sum to 1 on the test data if all the stimulus-related variance was reliable in the generalization test.

**4.2.3 Permutation tests for significance.** To determine the range of ranks where each spectrum is statistically significant, we perform permutation tests. Specifically, we follow the general formulation above exactly except for the second step, where the test data are projected onto the shared latent space. Instead of projecting $X_{\text{test}}$ and $Y_{\text{test}}$ onto the latent space using $U$ and $V$, we instead first permute the rows of $Y_{\text{test}}$ to shuffle the stimulus responses and obtain $Y_{\text{test}}^{\text{permuted}}$. We then compute the permuted spectrum

$$\Sigma^{\text{permuted}}(X, Y) = \text{cov}\left(X_{\text{test}} U, Y_{\text{test}}^{\text{permuted}} V\right) ,$$

which has a null distribution centered at 0 (Fig 3, inset). We repeat this process $N = 5,000$ times to obtain 5,000 permuted spectra and compute the 68th, 95th and 99th percentiles of the empirical null distribution, corresponding to the gray contour lines in Fig 3, which illustrate how many standard deviations away from zero the true spectrum lies.

**4.2.4 Relationship to other spectral estimators.** Note that the first step of the cross-decomposition procedure is a well-established statistical method also known as Procrustes transformation or partial least squares singular value decomposition (PLS-SVD). This alignment of two datasets based on the spectral decomposition of their cross-covariance is also used in the hyperalignment procedure [3], though [3] focus on aligning one dataset to another and not on the underlying spectrum of variance shared between the datasets. [10] developed cross-validated principal component analysis (cvPCA) to estimate such a spectrum of stimulus-related variance shared between repeated neural responses to the same stimuli. While the cross-decomposition approach shares similarities with the covariance eigenspectrum obtained by cross-validated PCA (cvPCA) [10], it also departs from it in two important ways. First, cvPCA can only be applied to within-subject analyses. To explore the level of shared representations between subjects, we need a method that can functionally align their representations—as cross-decomposition does. Second, cvPCA examines reliability across stimulus repetitions for a set of images but does not examine generalization to new images. Because our goal is to characterize representational properties that generalize over images, our cross-decomposition approach assesses both reliability across stimulus repetitions *and* generalization to new stimuli. As a result of these differences, cross-decomposition and cvPCA characterize different statistical quantities whose spectra naturally differ. This is illustrated in Fig 1, where standard PCA, cvPCA, and cross-decomposition exhibit different spectral decays due to their different generalization requirements. Note that an important consequence of this is that the power-law exponents observed for different estimators should not be

directly compared. Additionally, we introduce here a method for computing a between-subject spectral correlation coefficient, which characterizes the range of dimensions over which the representation in one subject is shared with another. This estimator is described in detail below.

### 4.3 Specific analyses

The general formalism described above applies to all the following analyses we perform, with variations only in the contents of the data matrices $X$ and $Y$.

**4.3.1 Computing within-subject spectra $\Sigma_k(X_1, X_2)$.** When evaluating the cross-validated covariance spectra for a single individual, say subject $X$, we compute the cross-covariance of neural responses to the same images on different trials, say $X_1$ and $X_2$. This ensures that the estimated covariance generalizes across repeated presentations of the stimuli.

**4.3.2 Computing between-subject or between-region spectra $\Sigma_k(X, Y)$ or $\Sigma_k(P, Q)$.** When computing *between-subject* spectra, $X_i$ contains neural responses from subject $X$ on the $i$th presentation of the shared stimuli while $Y_j$ contains neural responses from subject $Y$ on the $j$th presentation of the shared stimuli. Our goal is to estimate the spectrum of reliable variance $\hat{\Sigma}_k(X, Y)$ shared between the subjects $X$ and $Y$. Since the ordering of trials is arbitrary, we estimate a symmetric spectrum by averaging the spectra for two pairs of trials: (i) $X_1$ and $Y_2$ and (ii) $X_2$ and $Y_1$, making our final estimate

$$\hat{\Sigma}_k(X, Y) = \frac{1}{2} \left[ \hat{\Sigma}_k(X_1, Y_2) + \hat{\Sigma}_k(X_2, Y_1) \right] .$$

Similarly, we can compute *between-region* spectra within an individual across different trials. Let $P_i$ be neural responses from region $P$ on the $i$th presentation of the shared stimuli and $Q_j$ be neural responses from region $Q$ on the $j$th presentation of the shared stimuli. The *between-region* covariance is computed with the above formula, replacing $X, Y$ by $P, Q$.

**4.3.3 Computing between-subject spectral correlation coefficients $r_k(X, Y)$.** The absolute amplitude of the covariance spectrum is not meaningful since it depends on the total variance in the system. However, we can compute a normalized spectral correlation coefficient characterizing the similarity between two subjects $X$ and $Y$ by computing (i) their between-subject cross-covariance $\Sigma_k(X, Y)$ relative to (ii) their within-subject covariances $\Sigma_k(X_1, X_2)$ and $\Sigma_k(Y_1, Y_2)$. Both (i) and (ii) are computed as described in the previous sections. Then, we simply compute the spectral correlation as

$$r_k(X, Y) = \frac{\Sigma_k(X, Y)}{\Sigma_k(X_1, X_2)^{1/2} \Sigma_k(Y_1, Y_2)^{1/2}} .$$

**4.3.4 Functional vs anatomical alignment.** In the lower panels of Fig 2, we refer to the between-subject spectra described above as *functionally* aligned spectra since they are computed by decomposing a cross-covariance matrix that assumes no anatomical alignment between the voxels across individuals. If we assume that different subjects' cortical surfaces are in fact anatomically aligned, i.e. the voxels at the same location in different subjects have identical tuning properties, we would expect that the latent dimensions from one participant should generalize to another. Specifically, we would expect that the rotation matrices $U$ and $V$ corresponding to the first and second subjects respectively could be swapped without losing any information when estimating cross-validated covariance on the test data.

We thus compute *anatomical* spectra by projecting $X_{\text{test}}$ onto $V$ instead of $U$ and $Y_{\text{test}}$ onto $U$ instead of $V$ and computing the covariance of the resulting projections. Since this procedure requires that the columns $U$ and $V$ correspond to the same anatomically aligned voxels across both subjects, we map all the neural responses onto the standard 1 mm Montreal Neurological Institute (MNI) template using pre-computed transformations provided by the Natural Scenes Dataset [13], downsample them to isotropic 1.8 mm voxels that match the resolution of the original dataset, and restrict our analyses to the common voxels that are present in both individuals. In S3 Fig and Fig 4, both the *functional* and *anatomical* spectra are computed on these data to ensure that they are directly comparable.

## 4.4 Gabor filter bank comparison

We generate a Gabor filter bank using quadrature pairs of Gabor filters, each of which obeys Eq 2:

$$g(x, y; \sigma, \gamma, f, \theta, \psi) = \exp\left(-\frac{x'^2 + \gamma^2 y'^2}{2\sigma^2}\right)\cos\left(2\pi f x' + \psi\right) \tag{2}$$

where $x$ and $y$ denote the location of the center of the filter relative to the image; $x' = x\cos\theta + y\sin\theta$ and $y' = -x\sin\theta + y\cos\theta$; $\sigma$ and $\gamma$ denote the scale and aspect ratio of the Gaussian envelope; and $f$, $\theta$ and $\psi$ denote the spatial frequency, orientation, and phase of the sinusoid.

Following [48], the feature space of this Gabor filter bank is the concatenation of three sets of features: (i) simple cell responses from each "odd" filter (phase $\psi = 0$), (ii) simple cell responses from each "even" filter (phase $\psi = \pi/2$), and (iii) complex cell responses modelled as the sum of the squares of (i) and (ii), which captures the "energy" of each quadrature pair. We use the following parameters: 3 scales ($\sigma = 0.025, 0.050, 0.075$, where the square image has relative side length 1); 1 aspect ratio ($\gamma = 1$); 3 frequencies at each scale $\sigma$ ($f = 0.25\sigma, 0.5\sigma, 0.75\sigma$); 8 orientations $\theta$ uniformly spaced from $[0, \pi]$; and $8 \times 8 = 64$ spatial locations ($x,y$ with 8 uniformly spaced locations in $[-0.45, 0.45]$, where the image center is at $(0,0)$). This produces 4,608 quadrature pairs, leading to a total of 13,824 features in the feature space. As a preprocessing step, each of these features is z-scored across all 73,000 stimuli in the Natural Scenes Dataset.

**4.4.1 Within-subject analysis.** We use $L_2$-regularized linear regression to fit the response patterns of each voxel in an example subject (subject 1) using features from the Gabor filter bank as predictors. The data were split into two halves containing responses to 5,000 stimuli each, used as the training and test sets respectively. The optimal shrinkage parameter was estimated for each voxel using leave-one-out cross-validation on the training set (out of 30 values logarithmically sampled from $10^4$ to $10^8$). Then, we used cross-decomposition to compute the spectrum of reliable stimulus-related variance shared between (i) neural responses on the first presentation of the stimuli and (ii) neural responses on the second presentation of the stimuli predicted by the Gabor model (S4 Fig, top, gray). We compare these to the within-subject covariance spectra reported in Fig fig:visual-regions, which compare actual neural responses on the first and second presentations of the stimuli (S4 Fig, top, green). To highlight differences between these spectra, we plot the cumulative variance of these shared latent dimensions (as in Figure 2F of [10]).

**4.4.2 Between-subject analysis.** Again, we use $L_2$-regularized linear regression to fit the response patterns of each voxel in an example subject (subject 2) using features from the Gabor filter bank as predictors. The data were split into two halves containing responses to 500 stimuli each (half the number of shared stimuli between subjects 1 and 2), used as the training and test sets respectively. The optimal shrinkage parameter was estimated as described earlier and we used cross-decomposition to compute the spectrum of reliable stimulus-related variance shared between (i) the neural responses of subject 1 and (ii) the neural responses of subject 2 predicted using features from the Gabor filter bank (S4 Fig, bottom, gray). We compare these to the between-subject covariance spectra reported in Fig 5, which compare actual neural responses from the first and second subjects (S4 Fig, bottom, red). As before, we plot the cumulative variance of these shared latent dimensions.

## 4.5 Representational similarity analyses

Representational similarity matrices (RSMs) were constructed for each subject by computing Pearson correlations between patterns of fMRI responses to pairs of images and repeating this for all image pairs (using single trial-level responses from the "nsdgeneral" region of interest for each image, exactly as in the between-subject cross-decomposition analyses shown in the right panel of Fig 2). RSA correlations were obtained by correlating the RSMs for two subjects using either the linear (Pearson) or rank-order (Spearman) correlation coefficient. This procedure reflects a standard approach for comparing the representations of two individuals or an individual and a model and is representative of a broader class of variance-weighted similarity metrics that also includes regression-based encoding models [15,49].

Reduced-rank fMRI data ("low-D") was generated by applying principal component analysis to the fMRI response matrices and then reconstructing each matrix from its first 10 principal components (PCs). Estimates of the variability of the RSA correlations were computed using bootstrap resampling ($N = 5,000$ samples). For each bootstrap sample, 90% of the stimuli were subsampled without repetition before computing representational similarity matrices (RSMs).

### 4.6 Replication datasets

We replicated our between-subject results in two other large-scale datasets. First, we analyzed the human fMRI data from the THINGS-data collection [16], where 3 participants viewed the same 8,640 object images belonging to 720 categories, allowing us to perform between-subject cross-decomposition. Then, we replicated our findings in the THINGS ventral stream spiking dataset [17], where multi-unit activity (MUA) from 1,024 microelectrodes (implanted in V1, V4, and IT) was recorded from two macaques in response to over 22,248 natural images, allowing us to perform between-monkey cross-decomposition using the preprocessed MUA data.

### Supporting Information

**S1 Fig Spectral normalization** (Left) Within-subject cross-trial spectra were computed with different numbers of voxels sampled from the general region of interest. Increasing the number of voxels leads to higher variance. (Right) Normalizing the spectra by the number of voxels accounts for these differences. Open symbols denote data that are not significant at $p < 0.001$ (permutation tests, $N = 5000$).
(TIFF)

**S2 Fig Varying the number of stimuli used for cross-decomposition.** Within-individual covariance spectra for an example subject (subject 1) are computed using different numbers of stimulus images, ranging from 766 (the number of shared images seen by all participants in the experiments) to 10,000. We note that with fewer stimuli, reliable signal at high-ranks is not detectable. Open symbols denote data that are not significant at $p < 0.001$ (permutation tests, $N = 5000$).
(TIFF)

**S3 Fig Comparison between cross-spectra obtained using functional and anatomical alignment, relative to a reference subject 1.** The two measured spectra are normalized by a power law to visualize their behavior over multiple orders of magnitude. While anatomical alignment reveals only $\approx 10$ shared latent dimensions, functionally aligning subjects reveals many more shared latent dimensions. Open symbols denote points where the mean is less than three standard deviations above zero.
(TIFF)

**S4 Fig A Gabor filter bank cannot fully explain the observed high-dimensional structure of the neural population responses.** (Top) (Gray) The normalized cumulative shared variance between (i) neural responses on the first stimulus presentation and (ii) neural responses on the second stimulus presentation predicted by the Gabor model. (Green) The normalized cumulative shared variance between the neural responses to the first and second presentations of the stimuli (within-subject spectrum shown in the main manuscript). Data are shown for example subject 1. (Bottom) (Gray) The normalized cumulative shared variance between (i) neural responses for the first subject and (ii) neural responses for the second subject predicted by the Gabor model. (Red) The normalized cumulative shared variance between the neural responses of the first and second subjects (between-subject spectrum shown in the main manuscript). In all regions from V1 through V4, the Gabor model is consistently lower-dimensional than the neural data, as demonstrated by the more rapid increase in cumulative variance. Error bands denote standard deviations across 8 folds of cross-validation. Note that the between-subject analysis (bottom) uses 10 times fewer stimuli than the within-subject analysis (top) and thus has larger error bands.
(TIFF)

**S5 Fig Between-region covariance spectra demonstrate the sequential processing of information from V1 to V4.** Between-region covariance spectra for visual cortex regions V1 to V4 and a large frontal region containing voxels that are not modulated by visual stimuli. All pairwise comparisons between V1, V2, V3, V4 and this frontal region are shown. The systematic decrease in between-region covariance from V1-V1 to V1-V4 recapitulates the sequential processing of information from V1 to V4. Open symbols denote data that are not significant at $p < 0.001$ (permutation tests, $N = 5000$). (TIFF)

**S6 Fig Between-subject covariance spectra in the general visual region, relative to each subject.** These plots show the same analysis as in the upper right panel of Fig 2 but with each subject treated as the reference subject. The last panel shows the average between-subject spectrum across all $\binom{8}{2} = 28$ pairs of comparisons. Open symbols denote data that are not significant at $p < 0.001$ (permutation tests, $N = 5000$). (TIFF)

**S7 Fig Representational similarity in visual cortex for all pairs of subjects.** These plot shows the same analysis as in Fig 4 but with each subject treated as the reference subject. (TIFF)

**S8 Fig Examples of singular vectors as a function of rank** obtained from our within-subject cross-decomposition analysis for subject 1 displayed on the cortical surface. The typical spatial scale decreases monotonically with rank. (TIFF)

**S9 Fig Visualizing images from the dataset that strongly activate different latent dimensions of V1.** To interpret the latent dimensions of primary visual cortex (V1) responses to natural images, we visualize the images from the dataset that have the most positive (left) and most negative (right) projections on the left singular vectors extracted from within-subject cross-decomposition for an example subject (subject 1). We show images corresponding to the first few latent dimensions (1-5) followed by a small sample of high-rank dimensions (8, 16, 32, 256, 1024) throughout the spectrum. We find that the second latent dimension appears sensitive to airplanes/trains while the third latent dimension is activated strongly by flowers in vases, and the fourth is activated by clocktower-like buildings. However, we expect that these patterns are driven by low-level features such as spatial orientation that covary strongly with these semantic features in these images. Meanwhile, the high-rank dimensions show no interpretable patterns. Due to restrictive licensing, the original images have been replaced in this visualization with semantically similar images generated from the diffusion model CommonCanvas-XL-C based on human-generated captions [50]. Note that the perceptual details of these synthesized images differ from the original images. (TIFF)

**S10 Fig Visualizing images from the dataset that strongly activate different latent dimensions of the visual cortex.** We repeat the analysis described in S9 Fig, using the "general" visually responsive region instead of primary visual cortex (V1). Here, the first latent dimension appears to be correlated with animacy (S13 Fig, top, first pair of bars): it separates images with people (right) from images without people (left). The high-rank dimensions again show no interpretable patterns. Due to restrictive licensing, the original images have been replaced in this visualization with semantically similar images generated from the diffusion model CommonCanvas-XL-C based on human-generated captions [50]. Note that the perceptual details of these synthesized images differ from the original images. (TIFF)

**S11 Fig Power-law spectra of between-subject covariance are also found in the THINGS dataset of fMRI responses to object images.** These plots show between-subject covariance spectra of visual cortex responses in V1 through V4 and object-selective lateral occipital complex (LOC) from the THINGS-data fMRI dataset [16]. All spectra are

normalized to account for differences in the number of voxels across participants and averaged within bins of exponentially increasing width and across 8 folds of cross-validation. Error bars denote standard deviations across these 8 folds. Open symbols denote data that are not significant at $p < 0.001$ (permutation tests, $N = 5000$).
(TIFF)

**S12 Fig Power-law spectrum of between-monkey covariance in the THINGS ventral stream spiking dataset.** This plot shows the between-monkey covariance spectrum of visual cortex responses in V1, V4, and IT from the THINGS ventral stream spiking dataset [17]. The spectrum is averaged within bins of exponentially increasing width and across 8 folds of cross-validation. Error bars denote standard deviations across these 8 folds. Open symbols denote data that are not significant at $p < 0.001$ (permutation tests, $N = 5000$). Note that the error bars are small and not visible. This is because this dataset contains twenty times more stimuli than channels, and thus, the covariance estimates are highly consistent across splits of the data.
(TIFF)

**S13 Fig Semantic category information is distributed over the low-rank dimensions of neural activity in the "general" visually responsive region of cortex.** Information about the presence of people (top), food (middle) and text (bottom) in an image is distributed over the first 10 latent dimensions of neural activity in the "general" visually responsive region of cortex, identified through within-subject cross-decomposition in an example subject (subject 1). Images that either contain (colored) or do not contain (gray) these object categories are projected onto each of these dimensions and these score distributions were compared using Mann-Whitney U tests to identify significant differences (*, $p < 0.01$; **, $p < 0.001$; ***, $p < 0.0001$). Bars and error bars indicate the means of the score distributions and their standard errors.
(TIFF)

**S14 Fig Latent dimensions from the THINGS fMRI dataset are correlated with dimensions obtained from a large-scale behavioral experiment.** We perform between-subject cross-decomposition of the neural responses in the lateral occipital complex (LOC) from subjects 1 and 2 and project all the images onto the latent dimensions from subject 1. Then, we average these projections across all images within each of the 720 object categories and correlate each of these neural dimensions with each of the 66 interpretable dimensions that capture human object similarity judgments in a large-scale behavioral experiment [16]. (Top) We find correlations between the neural and behavioral dimensions but no evidence for a one-to-one correspondence between these dimensions. (Bottom) For each neural dimension, we compute the maximum absolute Pearson correlation with all behavioral dimensions and find that the correlations between neural and behavioral dimensions are strongest in the low-rank dimensions and fall off rapidly.
(TIFF)

## Author contributions

**Conceptualization:** Raj Magesh Gauthaman, Brice Ménard, Michael F. Bonner.

**Formal analysis:** Raj Magesh Gauthaman.

**Funding acquisition:** Brice Ménard, Michael F. Bonner.

**Investigation:** Raj Magesh Gauthaman.

**Methodology:** Raj Magesh Gauthaman, Brice Ménard, Michael F. Bonner.

**Project administration:** Brice Ménard, Michael F. Bonner.

**Software:** Raj Magesh Gauthaman.

**Supervision:** Brice Ménard, Michael F. Bonner.

**Validation:** Raj Magesh Gauthaman.

**Visualization:** Raj Magesh Gauthaman.

**Writing – original draft:** Raj Magesh Gauthaman.

**Writing – review & editing:** Raj Magesh Gauthaman, Brice Ménard, Michael F. Bonner.

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
