## [Decision Letter · Decision Letter 0]

8 Apr 2025

 PCOMPBIOL-D-25-00160

Universal scale-free representations in human visual cortex

PLOS Computational Biology

Dear Dr. Gauthaman,

Thank you for submitting your manuscript to PLOS Computational Biology. After careful consideration, we feel that it has merit but does not fully meet PLOS Computational Biology's publication criteria as it currently stands. Therefore, we invite you to submit a revised version of the manuscript that addresses the points raised during the review process.

Please submit your revised manuscript within 60 days Jun 07 2025 11:59PM. If you will need more time than this to complete your revisions, please reply to this message or contact the journal office at ploscompbiol@plos.org. Please include the following items when submitting your revised manuscript:

We look forward to receiving your revised manuscript.

Kind regards,

Tim Christian Kietzmann, Dr. rer. nat.

Academic Editor

PLOS Computational Biology

Daniele Marinazzo

Section Editor

PLOS Computational Biology

**Journal Requirements:**

4) We notice that your supplementary Figures are included in the manuscript file. Please remove them and upload them with the file type 'Supporting Information'. Please ensure that each Supporting Information file has a legend listed in the manuscript after the references list.

Potential Copyright Issues:

i) Please confirm (a) that you are the photographer of 1, or (b) provide written permission from the photographer to publish the photo(s) under our CC BY 4.0 license.

ii) Figures 1, 2, 5, and S6. Please confirm whether you drew the images / clip-art within the figure panels by hand. If you did not draw the images, please provide (a) a link to the source of the images or icons and their license / terms of use; or (b) written permission from the copyright holder to publish the images or icons under our CC BY 4.0 license. Alternatively, you may replace the images with open source alternatives. See these open source resources you may use to replace images / clip-art:

6) Thank you for stating "The dataset analysed in this study is the publicly available Natural Scenes Dataset (Allen et al., 2021)." Please provide a direct link to access the dataset. 

7) Please ensure that the funders and grant numbers match between the Financial Disclosure field and the Funding Information tab in your submission form. Note that the funders must be provided in the same order in both places as well.  Currently, the order of the funders is different in both places.

**Reviewers' comments:**

Reviewer's Responses to Questions

Reviewer #1: This paper by Raj and colleagues analyzes the structure of visual (fMRI) representations and show that the variance decays as a power law. This suggests that there is no systematic scale that dominates representations (information is distributed across dimensions). The study goes beyond prior approaches that employed statistical methods like PCA and ICA in characterizing the statistical structure (and not focusing on the specific visual feature encodings supported by the dimensions). Additional key results also demonstrate that A) the scale-free structure is remarkably preserved across individuals (Figure 2,4), B) is not observed in non-visual frontal regions (when compared between-subjects, Figure 3), and C) is present across the visual hierarchy (until V4, Figure 5,6). Overall, the paper is well-written, with clear results and a fair consideration of alternative explanations, including exponential decay and low-dimensional models.

I have three main concerns regarding the strong claim of “universality” in scale-free representations of the visual cortex.

1. Since the study does not use the full NSD stimulus set (due to cross-subject comparisons), it would be valuable to test whether these findings replicate in other large-scale fMRI datasets, such as THINGS or BOLD5000v2. Expanding the analysis to additional datasets would strengthen the claims about the universality of the scale-free structure and confirm that the results are not specific to the NSD dataset. There is a small concern of non-independence between train and test sets in NSD (as noted by Shirakawa et al., 2024). This issue could be completely mitigated by verifying whether the same statistical properties hold in an entirely separate dataset. A cross-dataset validation would provide stronger evidence that the observed scale-free structure is a robust feature of visual cortex organization rather than an artifact of dataset-specific factors.

2. At some stage, neural representations must become low-dimensional to align with behaviorally relevant features. A key question is whether the scale-free structure observed here is specific to early and mid-level visual areas or if it extends to higher-level visual regions. Since the study primarily focuses on early-to-mid-level regions, it’s unclear whether the same pattern would hold in later stages of processing.

One way to test this is by analyzing artificial neural networks (ANNs). If the findings generalize, we would expect early and mid-level layers to exhibit scale-free structure, while higher layers—where abstract, behaviorally relevant representations emerge—would not. Testing this would provide stronger support for claims of universality and clarify whether this pattern is truly a fundamental property of the visual cortex or just a characteristic of its early processing stages.

3. How much of the observed result is because of fMRI-related smoothing? Since fMRI signals aggregate activity across thousands of neurons within each voxel, it’s possible that this averaging inherently produces a power-law appearance in the variance spectrum.

The analyses of frontal regions addresses this problem to some degree. Al alternate strategy could be to conduct simulations to test the conditions under which scale-free structure could emerge simply from averaging over many independent neural responses. Another option is compare the same results against publicly available monkey Utah array recordings (like the Many Monkeys dataset from the DiCarlo lab).

Reviewer #2: This manuscript studies the dimensionality of visual representations across subjects in responses to naturalistic images. Rather than seek to understand what labeled, computed, or otherwise described visual features or dimensions of features drive responses in different functionally or anatomically labeled areas, the authors seek to understand the distribution of dimensions of the visual representation in terms of (1) how many dimensions of visual responses are shared across individuals, (2) how much variance each (range) of those dimensions explains. They find that the variance explained by stimulus-driven dimensions of visual features decreases in a way consistent with scale-free representation of visual information.

This paper is technically sound (tho with some omissions, see below), and takes a somewhat novel approach to understanding visual representations. My main issue with this manuscript is that I have a hard time understanding what the results add to our understanding of the visual system. I do not want to disparage the approach for being unfamiliar to me, and the analyses seem well-constructed and valid. However, I think the authors need to do a better job explaining how their results connect to extant literature, and need to do a better job explaining their dependent measures and how to interpret them.

Major points:

1. It is not clear to me that the observation of a scale-free representation is unexpected or otherwise surprising. Should we expect representation of visual information to be scale-free, either a priori (based on first principles) or based on other work? It sound from the literature cited as if the weight of available evidence - prior to this paper - suggests a scale-free representation. So what does this manuscript add to the insights derived from other papers? It would help if the authors framed the implications of their results in a way that did not use the words "scale-free" as much. The repetition of this phrase makes the results seem abstruse and/or tightly linked to this specific analysis. What were meaningful alternatives that may have been encountered here, and what would they have meant?

2. The authors need to do a better job connecting their results to the extant literature on dimensions of visual representation. They neither demonstrate nor speculate on what visual information may be in either the low-rank or the high-rank dimensions. There is a substantial literature (which the authors cite) on what the first few dimensions (generally up to 3-4, but as many as tens in some cases) of visual representation may contain, e.g. an inanimate/animate distinction, a fovea-periphery distinction, a real-world-size distinction, differences between spiky and smooth stimuli and other object properties, and more - Bao et al., 2020; Contier et al., 2024; Hebart et al., 2023; Huth et al., 2012; Tarhan & Konkle, 2020). Are there any indications that the low-rank dimensions span a similar space as any of these proposed dimensions?

3. The authors should also at least speculate on, and ideally show, what information is or may be contained in the higher dimensions that they report here (as these are potentially the most novel observation, to my eye). Could they reflect retinotopic position selectivity? information about exemplar-level differences between objects? Nuances of texture or material properties? The authors critique brain mapping studies as being fundamentally limited in the number of dimensions that can be visualized in brain maps. This is fair enough; however, a description of what features the dimensions capture need not be limited to brain maps per dimension or direct interpretations of each dimension; what space does the neural activity span? what distinctions among stimuli or brain states can be decoded from brain activity with low dimensions only, and with the additional dimensions revealed? To me, the lack of (any) interpretation of the dimensions in this manuscript puts a strong limit on the insights offered by the manuscript.

4. It's not clear that the highest-rank dimensions (beyond 100 dimensions or so) are reliably found across subjects. As far as I can see, the authors only perform significance tests for one pair of subjects (in Fig 3), and in that figure it's not totally clear that the highest-rank bins of dimensions explain reliably more variance than chance. (Perhaps due to the logarithmic Y axis, the error bars span the P cutoff values.) In figure S4, there seems to be a substantial falloff of variance explained beyond ~100 dimensions for many pairs of subjects. Also, the same calculation of chance in Fig 3 should be applied to all the subjects in figure S4 as well, and some summary analysis should be provided to assess reliability of high-rank dimensions across all subjects. What the authors do and do not consider to be reliable should be made very clear. I realize this will be computationally intensive, but this is an important point, because whether or not those higher-rank dimensions are reliable seems to be the difference between finding MORE dimensions than past studies and finding a truly scale-free representation.

Relatedly, it would help to discuss more how measurement noise affects the estimates of reliability of the highest-rank dimensions.

5. The authors raise the point that the number of dimensions revealed in the brain may depend on the stimulus set; they write: "The scale-free nature of these representations implies that their dimensionality is ill-defined: estimates of effective dimensionality for visual cortex likely reflect the properties of a given dataset such as the number and richness of stimuli or the number of recording channels rather than an intrinsic property of cortical representation." This argument would be more persuasive with a demonstration that other stimulus sets yield fewer dimensions, or a faster falloff of covariance over ranks. A good option for such an analysis would be functional localizer and/or PRF mapping stimuli included for the same participants in NSD. There are fewer unique TRs for these stimulus sets, but demonstrating that the elicited responses show a faster falloff of covariance by rank would make clear the degree to which the estimated dimensionality of the visual representation depends on the stimulus set.

6. Broadly speaking, it would be useful to describe in more intuitive terms what the dimensions they report are. As far as I understand, each dimension derived from the cross-decomposition is a pattern of responses across (visual) voxels for a particular subject. Similarly high values in different voxels for each dimension reflect commonalities in what visual factor or factors drives the voxels. Is this approximately correct? Some language to this effect (obviously corrected if I have misapprehended anything) would be helpful to the reader.

Minor comments:

In all cross-subject analyses, each dimension is found by decomposing the covariance matrix of a pair of subjects, and evaluated on held-out data between those same two subjects. Thus, there does not seem to be any guarantee that the dimensions are consistent beyond that pair. This, to me, means that the word 'universal' should be used with care. Referring to a "universal scale-free spectrum" seems fair, but a "universal scale-free code" seems to imply that a universal code or encoding has been found, which to me strongly implies that the same set of dimensions has been found for all subjects, which is not the case. So the phrase "universal scale-free code" seems to be overreaching the data.

In figure 3, it would be better to plot the mean of the null distribution rather than or in addition to the few examples of permuted distributions.

The graphs of spectral covariances and spectral correlations do not show values at every possible rank, but only at a sampling of ranks (at the centers (?) of bins of ranks). The authors write in the caption of Figure 2 that the covariances are "averaged within bins of exponentially increasing width", but this is the only information I could find on the bins. The authors should specify precisely what the bins are in the methods (and if these differ across subjects due to different numbers of stimuli presented and/or voxels, this should be clarified), and how the specific logarithmic spacing of bins were chosen.

It would be useful for the authors to give more intuition for how to think about the covariance values in many of their plots. The authors note that the values are not strictly positive, as within-set variance explained by PCs would be, but are they bounded in any way? Does the normalization applied to the covariance estimates make the estimated covariances cumulatively sum to 1 (or approximately 1)?

In the introduction, the authors should cite a reference for their estimate of the number of neurons in human visual cortex. "the effective dimensionality may not have an intrinsic upper bound other than the total number of neurons, which is about 109 in human visual cortex."

In the methods and the results, the authors state that approximately 1,000 images were shown to all subjects, and note that some subjects didn't see all images. The authors should be more precise. 907 images were shown to two participants; what was done about the other 93 images that these participants didn't see in the cross-subject decompositions? It seems that *n* (number of stimuli) has to be consistent for the math to work, so: Were only the common 907 images used in the cross-subject decompositions? were different numbers of images used for cross-decomposition in different pairs of subjects, each depending on the minimum number of common images between each pair of subjects? (I think so, based on what the authors have written, but this could be clearer.)

on p 13, the authors write "However, as we have shown, human brain representations are expressed using the full dimensionality of the available space (ultimately defined by the number of neurons)"

This seems to be reaching beyond the data; better to say "ultimately defined by the number of voxels" (or measurements made)

Reviewer #3: In this study, the authors analyze stimulus-related fMRI responses from the Natural Scenes Dataset (NSD) using spectral analysis of cross-validated cross-covariance. The main findings are that these responses exhibit high dimensionality and follow a scale-free (power-law) spectrum across four orders of magnitude. Furthermore, the authors show that many of these dimensions are shared across NSD participants, spanning approximately three orders of magnitude. This work complements previous findings in mouse V1 by Stringer et al. (2019, Nature), who similarly observed scale-free population activity patterns.

This is an interesting and timely study that is likely to attract considerable attention within the neuroscience community—and potentially spark debate. The paper is well written, clear, and engaging. The overall approach, which uses cross-validated spectral analysis of stimulus-related responses, is both straightforward and well motivated. However, given that the central claims regarding scale-free, high-dimensional representations rest entirely on the validity of the analyses (as the work is observational in nature), I believe that additional scrutiny prior to publication is warranted. Below, I outline several issues the authors may wish to address before the paper is published.

1. Given the spatial tuning properties of fMRI voxels, a certain degree of high dimensionality is expected. It would be beneficial to compare the observed dimensionality to that predicted by a linearized Gabor wavelet encoding model. How much of the observed dimensionality exceeds what would be expected from spatial tuning alone? Could the observed spectrum be fully explained by such a model?

2. The authors briefly acknowledge the assumption of linearity in their discussion but do not provide concrete controls for how violations of this assumption might affect their conclusions. Two plausible scenarios are not ruled out: (1) stimulus-related neural responses may lie in a low-dimensional nonlinear manifold, which PCA-based analyses would not capture (this possibility is mentioned in the discussion); and (2) stimulus-related neural responses may lie in a low-dimensional linear subspace that is distorted by nonlinearities in the hemodynamic response. In both cases, the assumption of linearity could lead to an overestimation of dimensionality. These concerns could be addressed through simulations quantifying their potential impact, through nonlinear analysis methods, and/or by comparison with spiking data.

3. While binning the spectrum improves statistical power and aids visualization, it also makes it more difficult to visually assess whether the spectrum truly follows a power-law (i.e., appears as a straight line in log-log coordinates). To strengthen the claim of scale-free structure, it would be helpful to conduct statistical model comparisons between a power-law fit and plausible alternatives. These comparisons should ideally be based on the unbinned spectral coefficients to avoid smoothing out potential deviations.

4. The NSD synthetic dataset has been very recently released. Performing measurements on synthetic stimuli could serve as an important test case for the generality of the authors’ claims, similarly to the usage of synthetic stimuli in Stringer et al., 2019.

5. It is somewhat unclear which methodological components are novel and which are adapted from previous work. Clarifying this distinction is important not only for assigning appropriate credit but also for identifying which aspects of the analysis may require further validation, for example through simulation studies. In particular, it would be helpful to clarify whether the correlation coefficient defined in Equation 1, used to quantify representational similarity across latent dimensions, is based on prior work or whether it is introduced here for the first time.

6. The treatment of RSA and related metrics is interesting, but it may benefit from a broader consideration of tradeoffs in representational comparison methods. While it is true that RSA, like other variance-weighted metrics, is largely insensitive to high-rank (i.e., low-variance) components, alternative approaches that do not apply variance weighting—such as canonical correlation analysis—have been shown to suffer from poor specificity (Kornblith et al., 2019 ICML). It may be useful to connect this work more explicitly to that literature and to consider the broader tradeoff between sensitivity to weak but potentially meaningful components and the risk of detecting spurious alignments.

**Have the authors made all data and (if applicable) computational code underlying the findings in their manuscript fully available?**

Reviewer #1: Yes

Reviewer #2: None

Reviewer #3: Yes

PLOS authors have the option to publish the peer review history of their article (what does this mean?). If published, this will include your full peer review and any attached files.

Reviewer #1: No

Reviewer #2: No

Reviewer #3: No

**Figure resubmission:**
---

## [Decision Letter · Decision Letter 1]

5 Aug 2025

PCOMPBIOL-D-25-00160R1

Universal scale-free representations in human visual cortex

PLOS Computational Biology

Dear Dr. Gauthaman,

Thank you for submitting your manuscript to PLOS Computational Biology. After careful consideration, we feel that it has merit but does not fully meet PLOS Computational Biology's publication criteria as it currently stands. Therefore, we invite you to submit a revised version of the manuscript that addresses the points raised during the review process.

Please submit your revised manuscript within 30 days Oct 05 2025 11:59PM. If you will need more time than this to complete your revisions, please reply to this message or contact the journal office at ploscompbiol@plos.org. Please include the following items when submitting your revised manuscript:

We look forward to receiving your revised manuscript.

Kind regards,

Tim Christian Kietzmann, Dr. rer. nat.

Academic Editor

PLOS Computational Biology

Daniele Marinazzo

Section Editor

PLOS Computational Biology

**Journal Requirements:**

1) Please note that Supplementary Figures should not be included in the manuscript. Only a full list of legends for your Supporting Information files should be added after the references list.

If the paper is accepted for publication, we will require that the Supporting Figures be removed from the main file of the manuscript as they only should be uploaded as separate files with the file type 'Supporting Information'

**Reviewers' comments:**

Reviewer's Responses to Questions

Reviewer #1: I thank the authors for their efforts in addressing my concerns. Their revisions have substantially clarified the paper, and I find the core claims more compelling.

However, I remain concerned that the calcium imaging analyses suffer from the same issues related to spatial smoothing as fMRI. To convincingly rule out these concerns, it would have been preferable to include a neurophysiology dataset less susceptible to spatial smoothing artifacts (such as the recently released NSD dataset from the Roelfsema Lab).

Reviewer #2: The authors have done a good job with the revision. The expanded discussion clarifies the relevance of the findings to the field, and they have added important methodological details (regarding number of images in the cross-covariance analyses and the definition of the bins) and statistical tests for the relevant plots. The analysis of how the reliability of the within-subject spectrum depends on the number of stimuli is a nice touch. The dimensionality analysis of the predictions of a fit Gabor model is also very nice. The authors have addressed most of the points I raised, but I have a small number of remaining questions and concerns which I am confident the authors can address.

1. Gabor dimensions & shared dimensions

As noted above, the demonstration that the predictions of a fit Gabor model are lower dimensional than the brain responses themselves is nice. However, I would be very curious to see whether the *shared* dimensions across participants are *also* higher dimensional than the Gabor model. It seems straightforward to make a similar plot for cumulative between-subjects covariance, or to add a line for between-subjects covariance to the same plot (fig S4). This would provide a useful test of whether the cross-subjects variance reported here is effectively subsumed by the Gabor model, and the rest of the within-subject variance is idiosyncratic to individuals. I don't view this as strictly necessary, but I think it is a relatively easy analysis to include and would strengthen the paper.

2. Interpretation of dimensions / relation to reported dimensions

The authors have provided visualizations of images at the ends of several dimensions they have uncovered. This is a partial answer to my question, but a more complete answer would relate the dimensions more directly to the dimensions reported in Khosla et al (2022) and Contier, Baker, & Hebart (2024). At least some of the dimensions appear to be similar.

3. The following statement is too strong:

The authors write: "our findings show that the shared representational dimensions of different brains can only be fully observed by hyperaligning the cortical activity patterns from one individual to another. This implies that the underlying dimensions of the representational code are not spatially localized but instead correspond to latent dimensions that are distributed."

Another strong possibility that should be acknowledge here is that the anatomical alignment (MNI space) was insufficient to capture local distortions or warpings of anatomy across individuals. Cortical alignment based on folds, function, and other anatomical properties has been shown to yield much better alignments than MNI space does. Thus, rejecting the possibility for anatomical consistency of high dimensions seems premature.

4. Richness of stimulus set

I think the way the authors have hedged to remove any discussion of the role of stimulus "richness" in estimating dimensionality is OK, but I do think it's obvious that stimulus richness must matter to estimates of dimensionality. To take an extreme case, a stimulus consisting of only black and white checkerboards flashed on and off would not be an effective stimulus for much of visual cortex and thus would not be likely to elicit the same dimensionality of responses as a large set of natural images. I think this is worth thinking about and potentially worth discussing, tho as I said the path the authors have taken is acceptable (if less interesting).

Reviewer #3: I thank the authors for their detailed rebuttal and extensive revisions. In particular, the Gabor-model analysis significantly strengthens the paper. All but two of the issues raised in my previous review have been sufficiently addressed. The two remaining issues are detailed below.

First, since statistical model comparisons between scale-free and alternative spectral distribution models were not conducted, the language regarding “scale-free” representations should be tempered. While straight lines on log-log plots are consistent with the authors’ claim, the inferential results (i.e., the permutation tests) only indicate that the representations are high-dimensional, not necessarily scale-free. The claim of “scale-free” representations appears throughout the paper, including in the title, abstract, and main text.

Second, my comment from the previous review remains largely unaddressed:

“It is somewhat unclear which methodological components are novel and which are adapted from previous work. Clarifying this distinction is important not only for assigning appropriate credit but also for identifying which aspects of the analysis may require further validation, for example through simulation studies. In particular, it would be helpful to clarify whether the correlation coefficient defined in Equation 1, used to quantify representational similarity across latent dimensions, is based on prior work or whether it is introduced here for the first time.”

As it stands, the Methods section (and especially subsections 5.2–5.3) does not clearly delineate which methods are adapted from previous research (particularly Stringer et al., 2019) and which are novel. This clarification should be made in the paper itself, not in the rebuttal.

Other than these two issues, I believe the paper is ready for publication.

**Have the authors made all data and (if applicable) computational code underlying the findings in their manuscript fully available?**

Reviewer #1: Yes

Reviewer #2: Yes

Reviewer #3: Yes

PLOS authors have the option to publish the peer review history of their article (what does this mean?). If published, this will include your full peer review and any attached files.

Reviewer #1: No

Reviewer #2: No

Reviewer #3: No

**Figure resubmission:**
---

## [Decision Letter · Decision Letter 2]

3 Nov 2025

PCOMPBIOL-D-25-00160R2

Universal scale-free representations in human visual cortex

PLOS Computational Biology

Dear Dr. Gauthaman,

Congratulations to a great paper. I would like to point you towards an outstanding (minor) issue raised by reviewer 3 (R3) about the abstract. I agree with their comment and would like to ask the authors to change the abstract as suggested. I hope that you can resubmit the paper shortly, upon which I will swiftly accept it for publication.

We look forward to receiving your revised manuscript.

Kind regards,

Tim Christian Kietzmann, Dr. rer. nat.

Academic Editor

PLOS Computational Biology

Daniele Marinazzo

Section Editor

PLOS Computational Biology

**Journal Requirements:**

We ask that a manuscript source file is provided at Revision. Please upload your manuscript file as a .doc, .docx, .rtf or .tex. If you are providing a .tex file, please upload it under the item type u2018LaTeX Source Fileu2019 and leave your .pdf version as the item type u2018Manuscriptu2019.

**Reviewers' comments:**

Reviewer's Responses to Questions

**Comments to the Authors:**

Reviewer #1: The authors's responses, particularly their replication with monkey physiology data, has addressed all my concerns. The paper is now significantly stronger than before. I am happy to recommend the paper for publication and congratulate the authors on these results.

Reviewer #2: The authors have done a very thorough job with the revisions, and have addressed all of my concerns very well. I congratulate them on a very nice piece of work.

Reviewer #3: I thank the authors for their detailed response. In particular, the additional contextualization of the methods will be useful to the community. I remain concerned about the strength of the overall framing, given the limitations. Nonetheless, the added paragraphs help highlight these issues, which may be further explored through post-publication discussion and future research. I have no further actionable suggestions for the main text.

Before publication, the authors may wish to temper two statements in the abstract:

1. "variance systematically decays as a power law": this account was not formally distinguished from alternative heavy-tailed models; consider qualifying it (e.g., "approximately") or otherwise tempering.

2. "This work reveals a new fundamental principle of neural coding in human visual cortex": this statement appears to elevate an observational result to a computational principle without evidence of mechanistic significance. The abstract would read more accurately if this claim were omitted.

**Have the authors made all data and (if applicable) computational code underlying the findings in their manuscript fully available?**

Reviewer #1: Yes

Reviewer #2: Yes

Reviewer #3: Yes

PLOS authors have the option to publish the peer review history of their article (what does this mean?). If published, this will include your full peer review and any attached files.

Reviewer #1: No

Reviewer #2: **Yes: **Mark D. Lescroart

Reviewer #3: No

**Figure resubmission:**
---

## [Editor Report · Decision Letter 3]

6 Nov 2025

Dear - Gauthaman,

We are pleased to inform you that your manuscript 'Universal scale-free representations in human visual cortex' has been provisionally accepted for publication in PLOS Computational Biology.

Best regards,

Tim Christian Kietzmann, Dr. rer. nat.

Academic Editor

PLOS Computational Biology

Daniele Marinazzo

Section Editor

PLOS Computational Biology

---

## [Editor Report · Acceptance letter]

PCOMPBIOL-D-25-00160R3

Universal scale-free representations in human visual cortex

Dear Dr Gauthaman,

I am pleased to inform you that your manuscript has been formally accepted for publication in PLOS Computational Biology. Your manuscript is now with our production department and you will be notified of the publication date in due course.

With kind regards,

Zsofia Freund
